# PIWI proteins tether the piRNA biogenesis machinery to mitochondria during mammalian spermatogenesis

Jie Gao[1,3], Canmei Chen[1,3], Guanyi Shang[1], Wenyang Yu[1], Ting Zhao[1], Yunfang Zhang [ID][1], Chen Chen[2] & Deqiang Ding [ID][1✉]

## Abstract

piRNA biogenesis occurs in the intermitochondrial cement (IMC) in mammalian germ cells. The mechanisms by which IMC components engage mitochondria to form an efficient piRNA biogenesis machinery remain elusive. Here, we demonstrate that PIWI proteins orchestrate the assembly and disassembly of the piRNA biogenesis machinery in mice. The mitochondrial-anchored protein ASZ1 specifically interacts with PIWIL2 and recruits PIWIL2 to IMC granules. Sequentially, piRNAs competitively bind PIWIL2, leading to ASZ1-PIWIL2 dissociation. In fetal male germ cells, ASZ1-PIWIL2-TDRD1 forms a seed complex to initiate the assembly of the piRNA biogenesis machinery. During postnatal meiosis, the TDRKH-PIWIL1-TDRD1 complex synergizes with the ASZ1-PIWIL2-TDRD1 complex to induce substantial IMC assembly and pachytene piRNA biogenesis through TDRD1-mediated phase separation. PIWI proteins act as bridges, tethering non-mitochondrial proteins to mitochondrial-anchored proteins in IMC granules with the assistance of TDRD1. Together, our findings establish the pivotal role of PIWI proteins in governing the spatiotemporal dynamics of piRNA biogenesis machinery during mammalian spermatogenesis.

**Keywords** PIWI-Interacting RNA; PIWI Protein; Germ Granule; Mitochondrion; Spermatogenesis
**Subject Categories** Development; RNA Biology

## Introduction

PIWI-interacting RNAs (piRNAs) are a class of germline-specific small regulatory RNAs that play evolutionarily conserved roles in safeguarding genome integrity and supporting germ cell development (Czech et al, 2018; Ozata et al, 2019; Wang et al, 2023). Mature piRNAs of 24–32 nt in length are processed from long single-stranded piRNA precursor transcripts through both primary processing and secondary amplification (Han et al, 2015; Li et al, 2013; Mohn et al, 2015). In mice, piRNA biogenesis is believed to occur within a specialized structure called intermitochondrial cement (IMC), an electron-dense germ granule located among clustered mitochondria in germ cells (Gao et al, 2024; Lehtiniemi and Kotaja, 2018). The piRNA biogenesis machinery, which is enriched in IMCs, comprises endonucleases, exonucleases, RNA helicases, scaffold proteins, and other regulatory factors, and its proper organization and coordination are essential for efficient piRNA biogenesis (Lehtiniemi and Kotaja, 2018; Olotu et al, 2023; Suyama and Kai, 2024; Wang et al, 2020). The assembly and disassembly of piRNA biogenesis machinery on mitochondria is a dynamic process to ensure continuous piRNA producing (Ge et al, 2019; Webster et al, 2015).

To date, five mitochondria-anchored proteins, TDRKH/TDRD2, ASZ1/GASZ, GPAT2, mitoPLD, and PNLDC1 have been identified, which play unique and essential roles in piRNA biogenesis (Ding et al, 2019; Ding et al, 2017; Ma et al, 2009; Saxe et al, 2013; Shiromoto et al, 2019; Watanabe et al, 2011). It's plausible to hypothesize that these mitochondria-anchored proteins serve as the "cornerstone" to recruit the non-mitochondrial piRNA processing factors into IMCs for piRNA biogenesis. For example, PIWIL1 is recruited to mitochondria via direct interaction with TDRKH (Ding et al, 2019; Wei et al, 2024). However, the mechanism of piRNA biogenesis machinery assembly remains largely unknown. It has been reported that TDRD1 exclusively localizes to IMCs and functions as a scaffold protein to promote IMC formation and piRNA biogenesis through phase separation (Chuma et al, 2006; Gao et al, 2024; Mathioudakis et al, 2012). Although TDRD1 provides the major driving force to trigger IMC assembly, how TDRD1 is recruited to mitochondria and organizes its "client proteins" on the outer mitochondrial membrane to form an efficient piRNA biogenesis machinery remains elusive.

PIWI proteins are the most important core factors for both piRNA biogenesis and function. Distinct PIWI orthologs associate with specific piRNA populations at various stages of germ cell development to execute stage-specific functions (Wang et al, 2023). In mice, three PIWI proteins (PIWIL1/MIWI, PIWIL2/MILI, and

[1]Shanghai Key Laboratory of Maternal Fetal Medicine, Clinical and Translational Research Center, Shanghai First Maternity and Infant Hospital, Frontier Science Center for Stem Cell Research, School of Life Sciences and Technology, Tongji University, Shanghai 200092, China. [2]Department of Animal Science, Michigan State University, East Lansing, MI 48824, USA. [3]These authors contributed equally: Jie Gao, Canmei Chen. ✉E-mail: dingdeqiang@tongji.edu.cn

PIWIL4/MIWI2) associate with distinct developmental stage-specific piRNA populations (Carmell et al, 2007; Deng and Lin, 2002; Kuramochi-Miyagawa et al, 2004). In fetal and neonatal gonocytes, PIWIL2 and PIWIL4 are known to associate with transposon sequence-rich fetal piRNAs to silence transposable elements and maintain germline genome integrity (Aravin et al, 2008; De Fazio et al, 2011; Wang et al, 2023). In adult male mice, PIWIL2 interacts with pre-pachytene piRNAs in germ cells at stages before pachytene spermatocytes. Subsequently, PIWIL1 and PIWIL2 associate with pachytene piRNAs, which are expressed starting from the pachytene stage of meiosis, to regulate meiotic and post-meiotic gene expression (Di Giacomo et al, 2013; Goh et al, 2015; Gou et al, 2014; Reuter et al, 2011; Vourekas et al, 2012). Interestingly, the IMCs are most prominent in pachytene spermatocytes, where they trigger extensive pachytene piRNA biogenesis (Gao et al, 2024). As central factors, the exact roles of PIWI proteins in IMC formation and piRNA biogenesis machinery assembly remain largely unknown. Notably, PIWIL2 is the only PIWI protein that associates with all piRNA population during mouse germ cell development. In both fetal gonocytes and pachytene spermatocytes, PIWIL2 is located at IMCs to participate in piRNA production. Subsequently, the PIWIL2-piRNA complex translocate from IMCs and ultimately localizes to the chromatoid body (CB), another type of germ granule in round spermatids (Aravin et al, 2009; Gao et al, 2024). Despite these insights, the mechanism underlying PIWIL2 spatial localization and the role of PIWIL2 in IMC assembly remain enigmatic.

Here, by combining cellular approaches and mouse genetics, we demonstrate that PIWI proteins govern the assembly and disassembly of piRNA biogenesis machinery during mouse spermatogenesis. Through systematic analysis of the interactions among piRNA pathway proteins, we identify two distinct protein complexes, ASZ1-PIWIL2-TDRD1 and TDRKH-PIWIL1-TDRD1, that form "seed complex" to trigger IMC assembly and piRNA biogenesis. The mitochondrial-anchored protein ASZ1 specifically interacts with PIWIL2, recruiting it to IMCs, where piRNAs subsequently binding to PIWIL2, promoting the disassociation of the ASZ1-PIWIL2-TDRD1 complex. Using postnatal germline conditional knockout mice, we further validate the physiological roles and interplay of PIWIL2, PIWIL1, and TDRD1 in adult germ cells. Together, our findings establish the pivotal role of PIWI proteins in tethering piRNA biogenesis machinery to mitochondria during spermatogenesis.

# Results

## ASZ1 specifically interacts with PIWIL2 and recruits PIWIL2 to mitochondria

How PIWIL2 is initially targeted to IMC to engage piRNA biogenesis is unclear. We hypothesize that PIWIL2 is recruited to mitochondria by directly interacting with mitochondrial-anchored proteins. To explore this, we employed a heterologous expression system in HeLa cells. HA-tagged PIWIL2 and GFP-tagged mitochondrial-anchored proteins, including ASZ1, mitoPLD, TDRKH, GPAT2, and PNLDC1, were co-expressed in HeLa cells. The results showed that PIWIL2 specifically colocalized with ASZ1

at mitochondria, while no such colocalization was observed with the other four mitochondrial-anchored proteins (Fig. 1A). We further performed co-immunoprecipitation (co-IP) in 293T cells and showed that PIWIL2 preferentially interacted with ASZ1 (Fig. 1B). These data suggest that PIWIL2 is recruited to IMCs by interacting with ASZ1.

To investigate the physiological protein interactomes of PIWIL2 and ASZ1 in germ cells, P20 mouse testes were used to perform immunoprecipitation using PIWIL2 or ASZ1 antibodies, followed by mass spectrometric (MS) analysis (Fig. EV1A). We identified a total of 76 PIWIL2-interacting proteins (fold change >4, $p < 0.05$). Among them, 12 piRNA pathway proteins were identified in PIWIL2 complexes, including MAEL, ADAD2, PIWIL1, MOV10L1, TDRD1, TDRD5, TDRD7, GPAT2, ASZ1, TDRKH, PNLDC1, and HENMT1 (Figs. 1C and EV1A) (Ozata et al, 2019; Xiong et al, 2023). This underlines PIWIL2 as a central player in piRNA biogenesis and function. On the other hand, 23 proteins were identified in the ASZ1 complex, which only included four piRNA pathway proteins: PIWIL2, MOV10L1, GPAT2, and TDRKH (Figs. 1D and EV1A). Moreover, TDRD1 and TDRD5 also showed minor enrichment in ASZ1 complex (Fig. 1D). We next sought to confirm the interaction of ASZ1 with non-mitochondrial piRNA pathway proteins present in ASZ1 IP-MS. Co-IP experiments in transfected 293 T cells revealed that ASZ1 exhibited a strong interaction with PIWIL2, but showed minimal or no binding to MOV10L1, TDRD1, and TDRD5 (Fig. EV1B). We therefore conclude that ASZ1 preferentially interacts with PIWIL2 and recruits PIWIL2 to mitochondria.

## PIWIL2 serves as an adapter to form the ASZ1-PIWIL2-TDRD1 complex

It has been reported that PIWIL2 is required for the mitochondrial localization of TDRD1, which promotes IMC assembly through phase separation (Gao et al, 2024; Vagin et al, 2009). We asked whether the ASZ1-PIWIL2 complex was enough to recruit TDRD1 to IMCs. PIWIL2 is known to directly associate with TDRD1 through its arginine methylation in the N-terminal RG/RA motif (Mathioudakis et al, 2012; Wang et al, 2009). Treatment with the arginine methylation inhibitor MTA significantly impaired PIWIL2-TDRD1 interaction but did not affect PIWIL2-ASZ1 interaction (Fig. EV1C–E). We next constructed a PIWIL2 arginine-to-lysine (RK) mutation in which all nine arginines were mutated to lysines in the N-terminal RG/RA motif and performed co-IP in 293T cells. PIWIL2-RK mutation dramatically disrupted PIWIL2-TDRD1 interaction (Fig. 1E) but did not reduce PIWIL2-ASZ1 interaction (Fig. 1F). These results suggest that PIWIL2 interacts with TDRD1 and ASZ1 through distinct mechanisms. We further examined the colocalization of PIWIL2 with TDRD1 and/or ASZ1 in transfected HeLa cells. PIWIL2 was recruited into cytoplasmic condensates or mitochondria after co-expressing with TDRD1 or ASZ1, respectively. However, when we co-expressed TDRD1, PIWIL2, and ASZ1 in HeLa cells, all three proteins were obviously enriched in the mitochondrial region (Fig. 1G). PIWIL2-RK mutation disrupted the colocalization of PIWIL2 with TDRD1, but did not disrupt the colocalization of PIWIL2 with ASZ1 (Fig. EV1F). We further co-expressed TDRD1, PIWIL2, and ASZ1 in HeLa cells and performed TEM. Notably, we observed

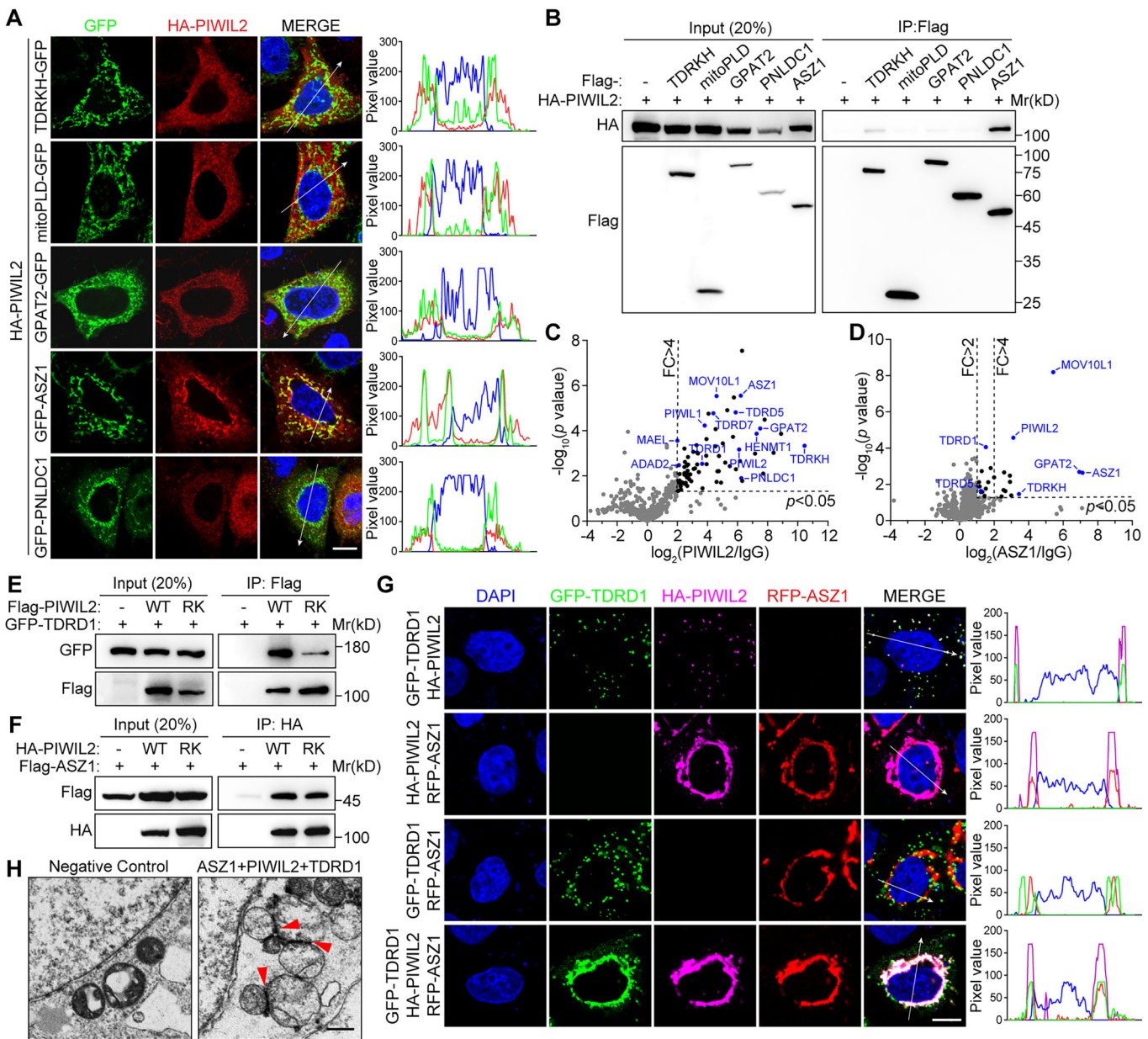

**Figure 1. PIWIL2 serves as an adapter to form the ASZ1-PIWIL2-TDRD1 complex.**

(A) HeLa cells were transfected with the indicated plasmids. Immunostaining was performed using the HA antibody. Scale bars, 10 μm. Fluorescence intensity through positions denoted by the white lines are shown in right. (B) Co-IP assay in 293T cells transfected with the indicated constructs. Flag-tagged and HA-tagged proteins were detected by WB. (C, D) Volcano plot showing enrichment and confidence of proteins identified by PIWIL2 (C) and ASZ1 (D) IP-MS from P20 mouse testes lysates. $n = 3$. Dotted lines indicate fold change (FC) >4, FC >2, and $p < 0.05$. The known piRNA pathway proteins are highlighted in blue. (E, F) Co-IP assay in 293T cells transfected with the indicated plasmids. Flag, GFP, and HA-tagged proteins were detected by WB. (G) HeLa cells were transfected with the indicated plasmids. Immunostaining was performed using the HA antibody. Scale bars, 10 μm. Fluorescence intensity through positions denoted by the white lines are shown in right. (H) Image of TEM on HeLa cells transfected with RFP-ASZ1, HA-PIWIL2, and GFP-TDRD1. The untransfected cells serve as a negative control. Mitochondrial-associated granules were indicated by red arrowheads. Scale bars, 1 μm. Data information: In (C, D), $p$ values were calculated using Student's $t$-test. The exact $p$ values for (C, D) are provided in Tables EV1 and EV2. Source data are available online for this figure.

mitochondrial-associated granules in transfected cells, resembling IMC structure in germ cells (Figs. 1H and EV1G). This suggests that ASZ1-PIWIL2-TDRD1 is sufficient to trigger IMC formation. Together, these results demonstrate that PIWIL2 simultaneously interacts with TDRD1 and ASZ1, serving as an adapter to form the ASZ1-PIWIL2-TDRD1 complex, which triggers IMC assembly.

## piRNA loading onto PIWIL2 disrupts ASZ1-PIWIL2 interaction

To further investigate how ASZ1 interacts with PIWIL2, we predicted ASZ1-PIWIL2 interaction using AlphaFold3 (Fig. EV2A,B) (Abramson et al, 2024). Base on the obtained models,

the N-terminal disordered regions of ASZ1 (around 1–40 aa) penetrated into and was anchored within a groove formed by the middle region of PIWIL2 (around 200–600 aa) (Fig. 2A). Both hydrophobic interactions and electrostatic interactions played roles in PIWIL2-ASZ1 interaction (Fig. 2B). To validate the predicted structural models, we constructed truncated ASZ1 and PIWIL2 to perform co-IP in 293 T cells. ASZ1 N-terminal deletion (ASZ1 Δ1–32) did not affect its mitochondrial localization (Fig. EV2C). Notably, ASZ1 N-terminal deletion (ASZ1 Δ1–32) or PIWIL2 middle region deletion (PIWIL2 Δ201–600) completely disrupted the ASZ1-PIWIL2 interaction (Fig. 2C,D). Moreover, PIWIL2 middle region (201–600 aa) rather than PIWIL2 N-terminal and C-terminal regions specifically interacted with ASZ1 (Fig. EV2D). Using the heterologous expression system in HeLa cells, we showed that ASZ1 Δ1–32 and PIWIL2 Δ201–600 disrupted the recruitment of PIWIL2 to mitochondria by ASZ1 (Fig. 2E). Together, our data confirm the ASZ1-PIWIL2 interaction model predicted by Alpha-Fold3, suggesting the pivotal role of the N-terminal region of ASZ1 and the middle region of PIWIL2 in mediating their interaction.

Interestingly, it is reported that the 3' end of piRNA is anchored within a binding pocket in the PAZ domain of PIWI proteins (Li et al, 2024; Matsumoto et al, 2016). When merging the published PIWIL2-piRNA structure and our predicted PIWIL2-ASZ1 structure, we noticed that the ASZ1 N-terminal motif was highly overlapped with the 3' end of piRNA (Figs. 2F and EV2E). We therefore speculated that the piRNA loading onto PIWIL2 may interfere with ASZ1-PIWIL2 interaction. To test this, we treated the adult testis lysates with RNase A and performed immunoprecipitation using PIWIL2 antibody (Fig. EV2F). RNase A treatment significantly enhanced ASZ1-PIWIL2 interaction, but not PIWIL2-TDRD1 interaction (Fig. 2G). We next performed ASZ1 immunoprecipitation in combination with RNase A treatment in adult testes. As expected, ASZ1-PIWIL2 interaction was augmented by RNase A treatment (Fig. 2H). We previously showed that PIWIL2 simultaneously interacts with TDRD1 and ASZ1, serving as an adapter to form the ASZ1-PIWIL2-TDRD1 complex. By TDRD1 immunoprecipitation and Western blotting (WB) in adult testes, TDRD1 interacts with comparable PIWIL2 but more ASZ1 after RNase A treatment (Fig. 2I). Together, these data demonstrate that piRNA competitively binding to PIWIL2 could disrupt PIWIL2-ASZ1 association, which may cause subsequent PIWIL2-piRNA translocation from mitochondria (Fig. 2J).

## TDRKH-PIWIL1 complex cooperates with ASZ1-PIWIL2 complex to recruit TDRD1 to mitochondria in pachytene spermatocytes

In adult testes, both PIWIL1 and PIWIL2 are located into IMCs to participate pachytene piRNA biogenesis. We next sought to investigate the combined role of PIWIL1 and PIWIL2 in IMC assembly during pachytene piRNA biogenesis. We performed TDRD1 IP-MS in P20 mouse testes to identify TDRD1-interacting proteins. Among a total of 77 identified TDRD1-interacting proteins, 12 were piRNA pathway proteins (Fig. 3A). PIWIL2 and PIWIL1 are the most abundant proteins in the TDRD1 complex (Fig. 3B). We therefore speculate that both PIWIL1 and PIWIL2 are involved in IMC formation by recruiting TDRD1 to mitochondria.

Co-IP assay in 293 T cell confirmed that TDRD1 indeed directly interacted with PIWIL1 (Fig. 3C). TDRD1 recognized arginine

methylated PIWIL2 through its Tudor domains (Wang et al, 2009). However, PIWIL1-TDRD1 interaction was unaffected after MTA treatment (Fig. EV3A). We next constructed a PIWIL1 RK mutation in which thirteen arginines were mutated to lysines in PIWIL1 N-terminal RG/RA motifs and revealed that PIWIL1 RK mutant sufficiently retained interaction with TDRD1 (Fig. 3C). To gain insight into the details of TDRD1-PIWIL1 interaction, we constructed truncated PIWIL1 proteins in which the N-terminal, middle region, and C-terminal were deleted separately. By co-IP in 293T cells, all truncated PIWIL1 proteins interacted with TDRD1, although with significantly lower intensity compared to the full-length protein (Fig. 3D). We next tested the TDRD1 interaction with individual PIWIL1 domains. Both the N-terminal region and PIWI domain interacted with TDRD1 (Fig. EV3B), suggesting that PIWIL1 engages with TDRD1 through multiple interfaces. TDRD1 formed cytoplasmic condensates autonomously through phase separation, which may enhance the recruitment of client proteins (Fig. EV3C). Co-IP results revealed that TDRD1 phase separation-deficient mutant (TDRD1-3GS) (Gao et al, 2024) significantly impaired PIWIL1-TDRD1 interaction (Fig. 3E). Together, these data indicate that TDRD1-PIWIL1 interaction is independent of the recognition module between the Tudor domain and arginine methylation but is enhanced by TDRD1 phase separation, highlighting the distinct mechanism by which TDRD1 recognizes different PIWI proteins.

PIWIL1 interacts with TDRKH through its N-terminal RG/RA motif (Fig. EV3D) (Wei et al, 2023). Based on this, we hypothesized that PIWIL1 simultaneously interacts with TDRD1 and TDRKH using different motifs and recruits TDRD1 to mitochondria. We detected the colocalization of PIWIL1 with TDRD1 and/or TDRKH in transfected HeLa cells. PIWIL1 was recruited into TDRD1 condensates or mitochondria after co-expressing with TDRD1 or TDRKH, respectively, in HeLa cells. Strikingly, when we co-expressed TDRD1, PIWIL1, and TDRKH in HeLa cells, all three proteins were obviously enriched in the mitochondrial region (Fig. 3F). PIWIL1 RK mutation disrupted the colocalization of PIWIL1 with TDRKH, but did not disrupt the colocalization of PIWIL1 with TDRD1 (Fig. EV3E). TDRKH-PIWIL1 complex failed to recruit TDRD1-3GS mutant to mitochondria (Fig. 3G). We further co-expressed TDRD1, PIWIL1, and TDRKH in HeLa cells and performed TEM. As expected, we observed IMC-like granules among mitochondria in transfected cells (Figs. 3H and EV3F). Since ASZ1 and PIWIL2 specifically recognized each other, we next examined the specificity of TDRKH-PIWIL1 interaction. By co-IP assay in 293T cells, we found that PIWIL1 specifically interacted with TDRKH but not GPAT2, mitoPLD, ASZ1, and PNLDC1 (Fig. EV3G). Consistently, only TDRKH was able to sufficiently recruit PIWIL1 to mitochondria in HeLa cells (Fig. EV3H). Taken together, these results demonstrate that PIWIL1 interacts simultaneously with TDRD1 and mitochondria-anchored TDRKH, recruiting TDRD1 to mitochondria to promote IMC formation.

## Loss of PIWIL2 does not disrupt IMC formation in pachytene spermatocytes

Since loss of PIWIL1 does not affect IMC formation in adult testes (Wei et al, 2023), we next investigate the physiological role of PIWIL2 in IMC formation in pachytene spermatocytes. We generated *Piwil2* conditional knockout (*Piwil2*^cKO^) mice in which *Piwil2* was deleted in spermatogonia starting at postnatal Day 3 by

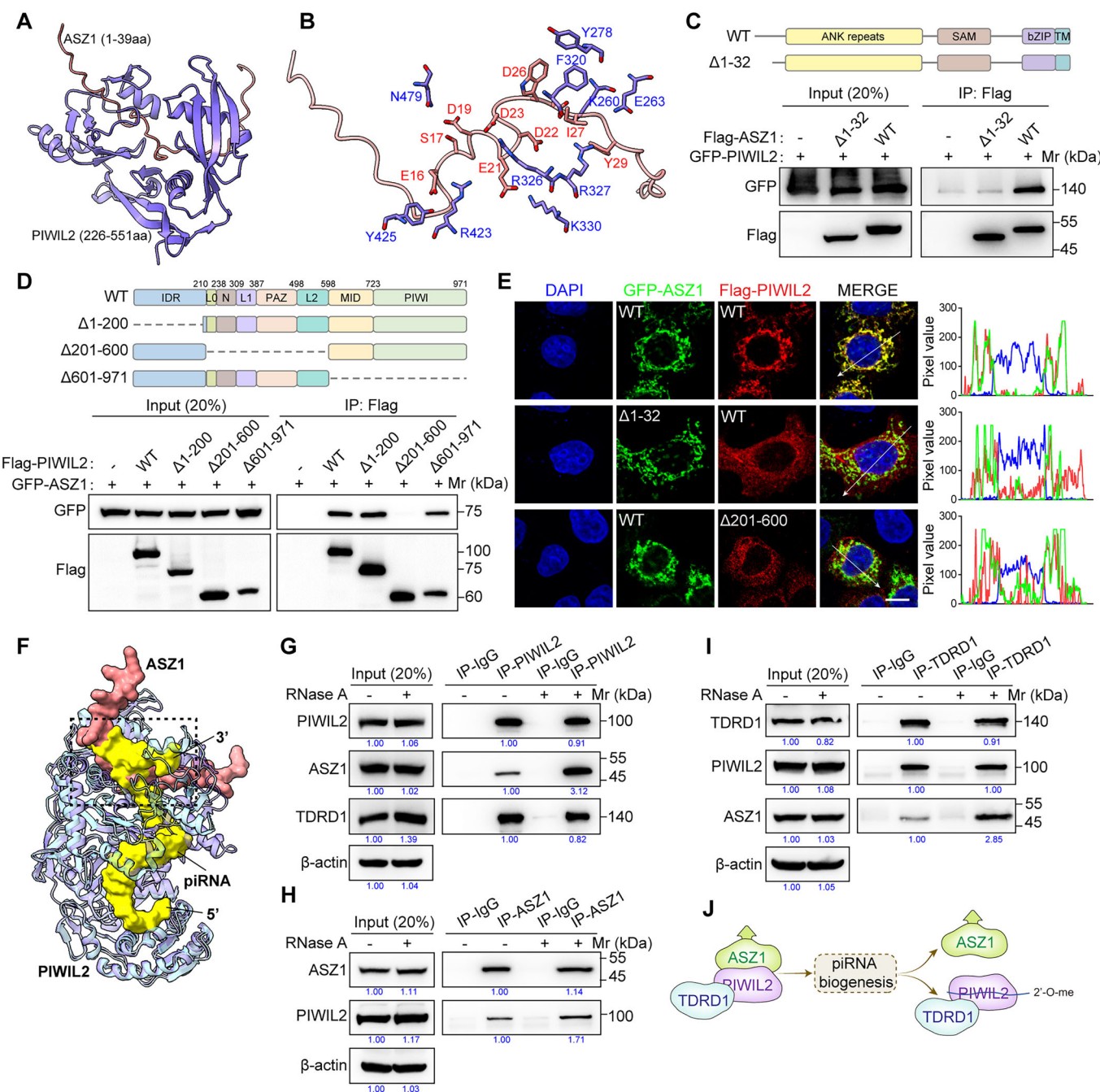

**Figure 2. piRNA loading onto PIWIL2 disrupts ASZ1-PIWIL2 interaction.**

(A) Interaction structure model of the ASZ1 N-terminal and PIWIL2 middle region predicted by AlphaFold3. (B) Molecular details of the interaction interface between the ASZ1 and PIWIL2. The interacting residues in ASZ1 (red) and PIWIL2 (blue) are labeled. (C, D) Co-IP and WB assay in 293T cells transfected with the indicated plasmids. Schematic representations of ASZ1 proteins (C) and PIWIL2 proteins (D) were shown on top. (E) HeLa cells were transfected with the indicated plasmids. Immunostaining was performed using the Flag antibody. Scale bars, 10 μm. Fluorescence intensity through positions denoted by the white lines are shown in right. (F) Comparison of interaction structural models for ASZ1-PIWIL2 complex and PIWIL2-piRNA complex. The PIWIL2-ASZ1 structure is predicted by AlphaFold3. PIWIL2-piRNA structure is from PDB (7YGN). piRNA is shown in yellow, and ASZ1 (1–39 aa) is shown in red. The dotted box highlights the interaction interface. (G–I) Co-IP assay of PIWIL2 (G), ASZ1 (H), and TDRD1 (I) in mouse adult testes with or without RNase A treatment. PIWIL2, ASZ1, and TDRD1 protein levels were detected by WB. (J) Schematic diagram showing that piRNA loading facilitates the dissociation of PIWIL2 and TDRD1 from ASZ1. Source data are available online for this figure.

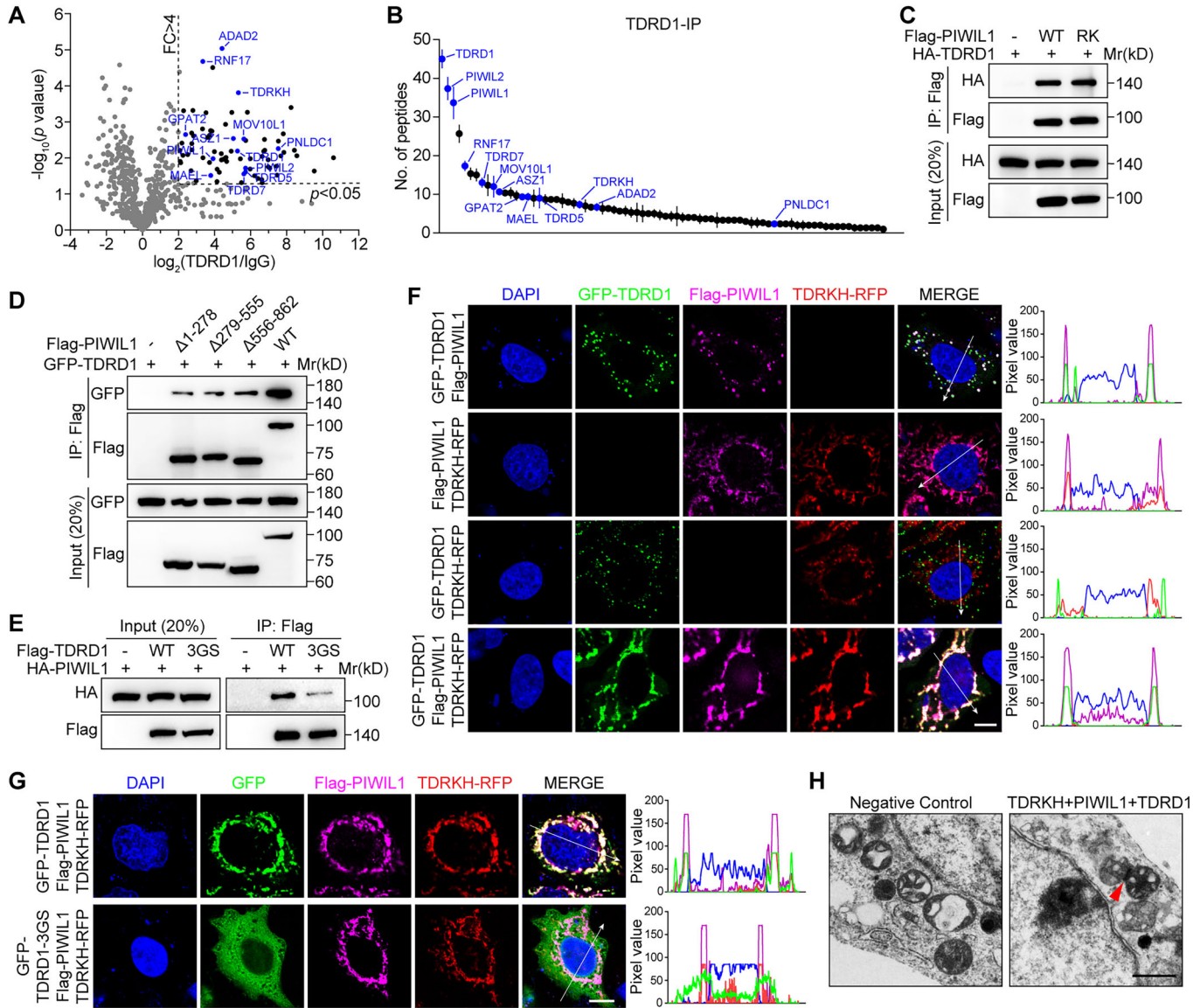

**Figure 3. TDRKH-PIWIL1 complex cooperates with ASZ1-PIWIL2 complex to recruit TDRD1 to mitochondria.**

(A) Volcano plot showing enrichment and confidence of proteins identified by TDRD1 IP-MS from P20 mouse testes lysates. $n = 3$. Dotted lines indicate FC >4 and $p < 0.05$. The known piRNA pathway proteins are highlighted in blue. (B) Scatter plot of the abundance of enriched proteins from TDRD1 IP/MS (FC >4 and $p < 0.05$ in A). The dots were arranged by peptide counts. $n = 3$. The known piRNA pathway proteins are highlighted in blue. (C–E) Co-IP and WB assay in 293T cells transfected with indicated constructs. Flag, GFP and HA-tagged proteins were detected by WB. (F, G) HeLa cells were transfected with indicated constructs. Immunostaining was performed using the Flag antibody. Scale bars, 10 μm. Fluorescence intensity through positions denoted by the white lines are shown in right. (H) Image of TEM on HeLa cells transfected with TDRKH-RFP, Flag-PIWIL1, and GFP-TDRD1. The untransfected cells serve as a negative control. Mitochondrial-associated granules were indicated by red arrowheads. Scale bars, 1 μm. Data information: In (B), data were presented as mean ± s.e.m. In (A, B), $p$ values were calculated using Student's $t$-test. The exact $p$ values for (A, B) are provided in Table EV3. Source data are available online for this figure.

Stra8-Cre (Fig. EV4A) (Chen et al, 2018). Immunostaining and WB using PIWIL2 antibody confirmed the successful ablation of PIWIL2 in *Piwil2^{cKO}* adult testes (Fig. EV4B,C). As previously reported, *Piwil2^{cKO}* mice exhibited smaller testes than control mice (Fig. EV4D) (Di Giacomo et al, 2013). Hematoxylin-eosin (H&E) staining and ACRV1 (an acrosomal marker) immunostaining revealed that the germ cells were primarily arrested at early stages of elongated spermatids (Fig. EV4E,F). TUNEL assay revealed that arrested germ cells underwent apoptosis in *Piwil2^{cKO}* testes (Fig.

EV4G). No mature sperm were observed in *Piwil2^{cKO}* epididymides (Fig. EV4H).

To examine pachytene piRNA biogenesis in *Piwil2^{cKO}* mice, we sequenced the total small RNA from adult testes. As expected, PIWIL2-bound piRNAs around 26 nt were completely absent in *Piwil2^{cKO}* testes (Fig. EV4I). Remarkably, PIWIL1-bound piRNAs around 30 nt were also significantly reduced in *Piwil2^{cKO}* testes, indicating that loss of PIWIL2 also impairs PIWIL1-bound piRNA (Fig. EV4I). This was further confirmed by WB that the PIWIL1

protein level was lower in *Piwil2cKO* testes (Fig. EV4C). We mapped the piRNA reads to the mouse genome and revealed that the majority of pachytene piRNAs were derived from piRNA clusters in both control and *Piwil2cKO* mice (Fig. EV4J).

We next investigated the role of PIWIL2 in IMC assembly and TDRD1 recruitment during pachytene piRNA biogenesis. By performing co-immunostaining, TDRD1 retained complete colocalization with TDRKH in *Piwil2cKO* pachytene spermatocytes (Fig. 4A). Loss of PIWIL2 did not affect the colocalization of TDRD1 with PIWIL1 (Fig. EV4K). Transmission electron microscopy (TEM) assay uncovered that the IMC structures were adequately present in both control and *Piwil2cKO* pachytene spermatocytes (Fig. 4B). This contrasted with *Piwil2−/−* gonocytes, which displayed complete absence of IMCs after the loss of PIWIL2 (Fig. EV4L). Interestingly, we observed mild mitochondrial aggregation in *Piwil2cKO* pachytene spermatocytes and incomplete CB formation in *Piwil2cKO* round spermatids (Figs. 4A and EV4M), which were frequently observed in various piRNA biogenesis-deficient mutant mice (Ding et al, 2019; Wei et al, 2023). Together, we conclude that PIWIL2 plays essential roles in pachytene piRNA biogenesis, while loss of PIWIL2 does not significantly disrupt TDRD1 recruitment and IMC formation.

## PIWIL2 preferentially binds to TDRD1 and regulates its stability in adult testes

We next investigated the physiological importance of PIWIL2-TDRD1 and PIWIL1-TDRD1 interactions. TDRD1 was sufficiently recruited to mitochondria in both *Piwil2cKO* and *Piwil1−/−* pachytene spermatocytes (Fig. 4A). Loss of PIWIL1 did not affect the colocalization of TDRD1 with PIWIL2 (Fig. EV4N). Strikingly, we observed that TDRD1 gradually degraded from late pachytene spermatocytes to round spermatids in *Piwil2cKO* adult testes (Fig. 4C). Consistently with this, the protein level of TDRD1 was decreased in *Piwil2cKO* adult testes (Fig. 4D). In contrast, immunostaining revealed that the expression level and localization of TDRD1 were largely unaffected in *Piwil1−/−* spermatocytes (Fig. 4C). We even observed stronger immunostaining signal of TDRD1 in CBs in *Piwil1−/−* round spermatids (Fig. 4C). WB results showed that the protein levels of PIWIL2 and TDRD1 were up-regulated in *Piwil1−/−* adult testes (Fig. 4E), indicating the compensatory role of PIWIL2 in *Piwil1−/−* mice. These data indicate that PIWIL2, rather than PIWIL1, is critical for TDRD1 stability in adult testes.

We next investigated the preference of PIWIL2-TDRD1 and PIWIL1-TDRD1 interaction. Co-IP assays in 293T cells revealed that PIWIL2 interacted more strongly with TDRD1 than PIWIL1 (Fig. 4F). To simultaneously detect the interaction of TDRD1 with PIWIL2 and PIWIL1, we employed a mitochondrial-anchored PIWIL2-TM and PIWIL1-TM recombinant protein in which PIWIL2 and PIWIL1 were fused with a transmembrane domain (TM) derived from ASZ1 (Gao et al, 2024). Both mitochondrial-anchored PIWIL1 and PIWIL2 successfully recruit TDRD1 to mitochondria (Figs. 4G and EV4O). PIWIL1 and PIWIL2 failed to recruit each other to mitochondria (Figs. 4G and EV4O). When we co-transfected cells with GFP-PIWIL1-TM, Flag-TDRD1, and HA-PIWIL2, both TDRD1 and PIWIL2 were recruited to mitochondria, indicating that TDRD1-PIWIL2 interaction is strong enough for their mitochondrial localization. On the other hand, when we co-transfected cells with GFP-PIWIL2-TM, Flag-TDRD1, and HA-

PIWIL1, only TDRD1 were recruited to mitochondria. The majority of PIWIL1 are distributed in the cytoplasm, indicating that TDRD1-PIWIL1 interaction is not enough for PIWIL1 mitochondrial localization (Figs. 4G and EV4O). Together, these data indicated that TDRD1 interacts more strongly with PIWIL2 than with PIWIL1.

## Mitochondrial-anchored proteins preferentially recognize PIWI proteins in piRNA biogenesis machinery

To assess the specificity of PIWI recruitment by TDRKH and ASZ1, we then systematically examined the recruitment of non-mitochondrial piRNA pathway proteins by mitochondrial-anchored proteins TDRKH, ASZ1, GPAT2, and mitoPLD in HeLa cells (Appendix Fig. S1A). PNLDC1 was not included because its own mitochondrial localization is largely dependent on TDRKH (Ding et al, 2019). Strikingly, among 19 tested non-mitochondrial piRNA pathway proteins, PIWIL2 is the sole protein that is sufficiently recruited to mitochondria by ASZ1 (Fig. 5A). On the other hand, TDRKH preferentially recognizes and recruits the other two PIWI proteins: PIWIL1 and PIWIL4 to mitochondria (Fig. 5B). The recruitment of PIWIL4 by TDRKH indicates that TDRKH might be involved in PIWIL4-bound piRNA biogenesis. As a comparison, GPAT2 and mitoPLD failed to obviously recruit any of the tested piRNA pathway proteins to mitochondria (Appendix Fig. S1B,C). These data suggest that mitochondrial-anchored proteins preferentially recognize PIWI proteins in piRNA biogenesis machinery.

TDRD1 forms condensates by phase separation, making it an ideal scaffold protein to recruit "client proteins" and trigger IMC assembly. Indeed, we detected many piRNA pathway proteins from TDRD1 IP-MS in mouse testes (Fig. 3A). To verify this, we examined the potential recruitment of piRNA pathway proteins by TDRD1 in HeLa cells. The results revealed that PIWIL2, PIWIL1, PIWIL4, and TDRD5 showed nearly complete colocalization with TDRD1 condensates, while MAEL, MOV10L1, MVH, FKBP6, EXD1, RNF17, TDRD6, TDRD12, and ADAD2 showed partial colocalization with TDRD1 condensates (Fig. 5C). This suggests that TDRD1 is critical for the recruitment and organization of IMC components, further highlighting the pivotal role of ASZ1-PIWIL2-TDRD1 and TDRKH-PIWIL1-TDRD1 complex in piRNA biogenesis machinery assembly.

## TDRD1 plays irreplaceable roles in IMC assembly and pachytene piRNA biogenesis in adult testes

PIWIL1 and PIWIL2 substitute for each other to sustain IMC formation and TDRD1 recruitment during pachytene piRNA biogenesis in adult testes. We wondered the necessity of TDRD1 in IMC formation and pachytene piRNA biogenesis. Considering that the germ cells were arrested before the pachytene stage in *Tdrd1−/−* mice (Gao et al, 2024), we generated *Tdrd1* conditional knockout (*Tdrd1cKO*) mice by crossing *Tdrd1 flox* mice with *Stra8*-GFP-Cre mice (Appendix Fig. S2A,B). *Tdrd1cKO* mice exhibited smaller testes than control mice (Fig. 6A). H&E staining revealed that the germ cells were arrested at round spermatids in *Tdrd1cKO* adult testes (Fig. 6B). No mature sperms were observed in *Tdrd1cKO* epididymides (Appendix Fig. S2C). ACRV1 staining revealed that the integrity of acrosomes was compromised starting from step 7-8

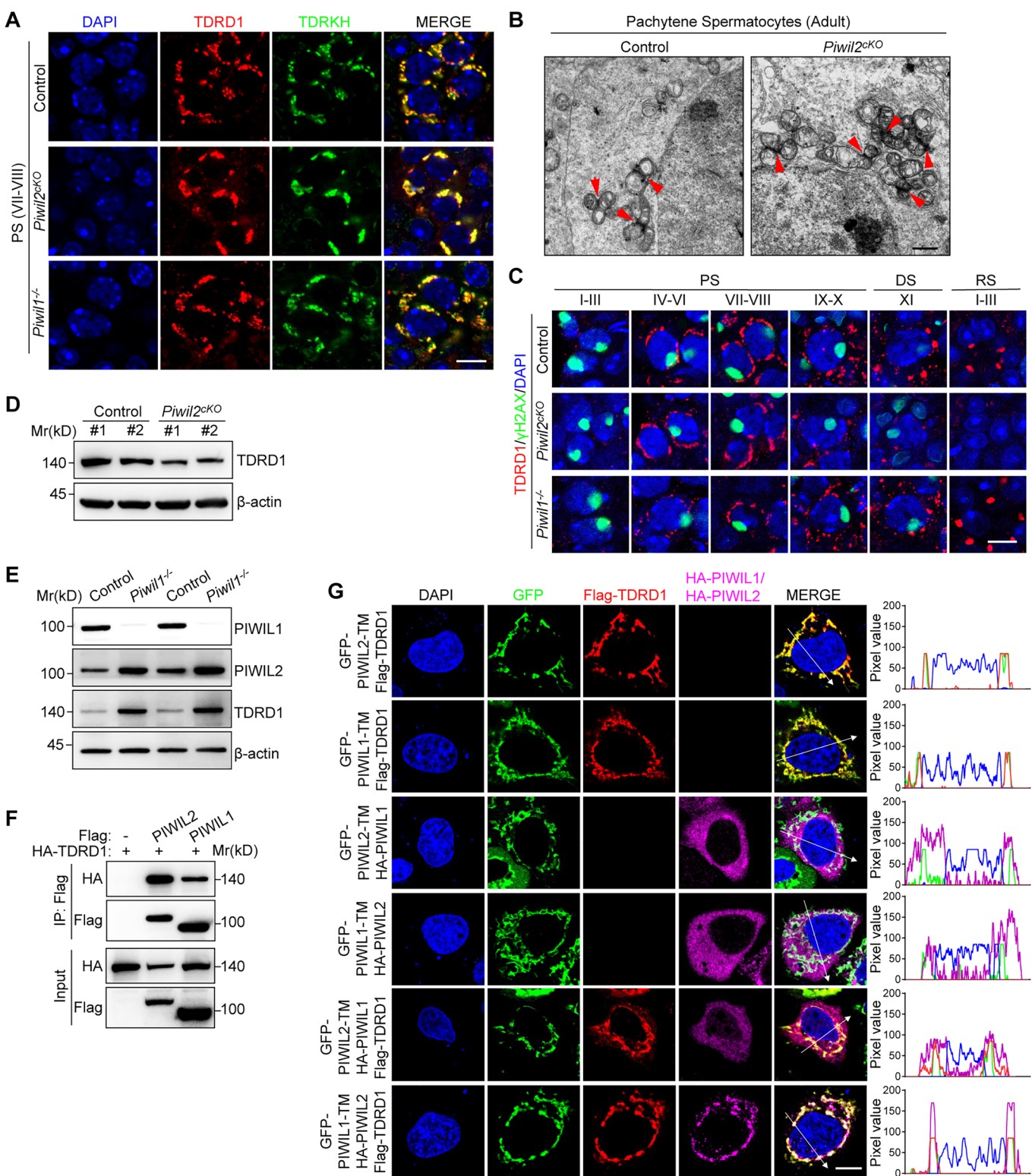

round spermatids in *Tdrd1^cKO* testes (Appendix Fig. S2D). TUNEL assay revealed that the arrested round spermatids underwent apoptosis in *Tdrd1^cKO* testes (Appendix Fig. S2E). Together, these data demonstrate that TDRD1 is essential for spermiogenesis in mice.

We next examined the IMC formation in *Tdrd1^cKO* mice. Immunostaining of the mitochondrial proteins TDRKH and ASZ1 in adult testes revealed that the mitochondria became more diffusely distributed in the cytoplasm of *Tdrd1^cKO* pachytene spermatocytes (Fig. 6C; Appendix Fig. S2F). Consistent with this,

**Figure 4. PIWIL2 preferentially binds to TDRD1 and regulates its stability in adult testes.**

(A) Co-immunostaining of TDRD1 and TDRKH on control, *Piwil2cKO* and *Piwil1−/−* adult testes. Stage VII–VIII seminiferous tubules were distinguished according to DAPI staining. PS, pachytene spermatocytes. Scale bars, 10 μm. (B) Images of TEM on pachytene spermatocytes from adult testes. IMCs were indicated by red arrowheads. Scale bars, 1 μm. (C) Co-immunostaining of TDRD1 and γH2AX on adult testes. PS pachytene spermatocytes, DS diplotene spermatocytes, RS round spermatid. Scale bars, 10 μm. The developmental stages of germ cells were distinguished according to γH2AX and DAPI staining. (D) WB of TDRD1 in control and *Piwil2cKO* adult testes. β-actin served as a control. (E) WB of PIWIL1, PIWIL2 and TDRD1 in control and *Piwil1−/−* adult testes. β-actin served as a control. (F) Co-IP and WB assay in 293T cells transfected with indicated constructs. Flag and HA-tagged proteins were detected by WB. (G) HeLa cells were transfected with indicated constructs. Immunostaining was performed using Flag, PIWIL1 and PIWIL2 antibodies. Scale bars, 10 μm. Fluorescence intensity through positions denoted by the white lines are shown in right. Source data are available online for this figure.

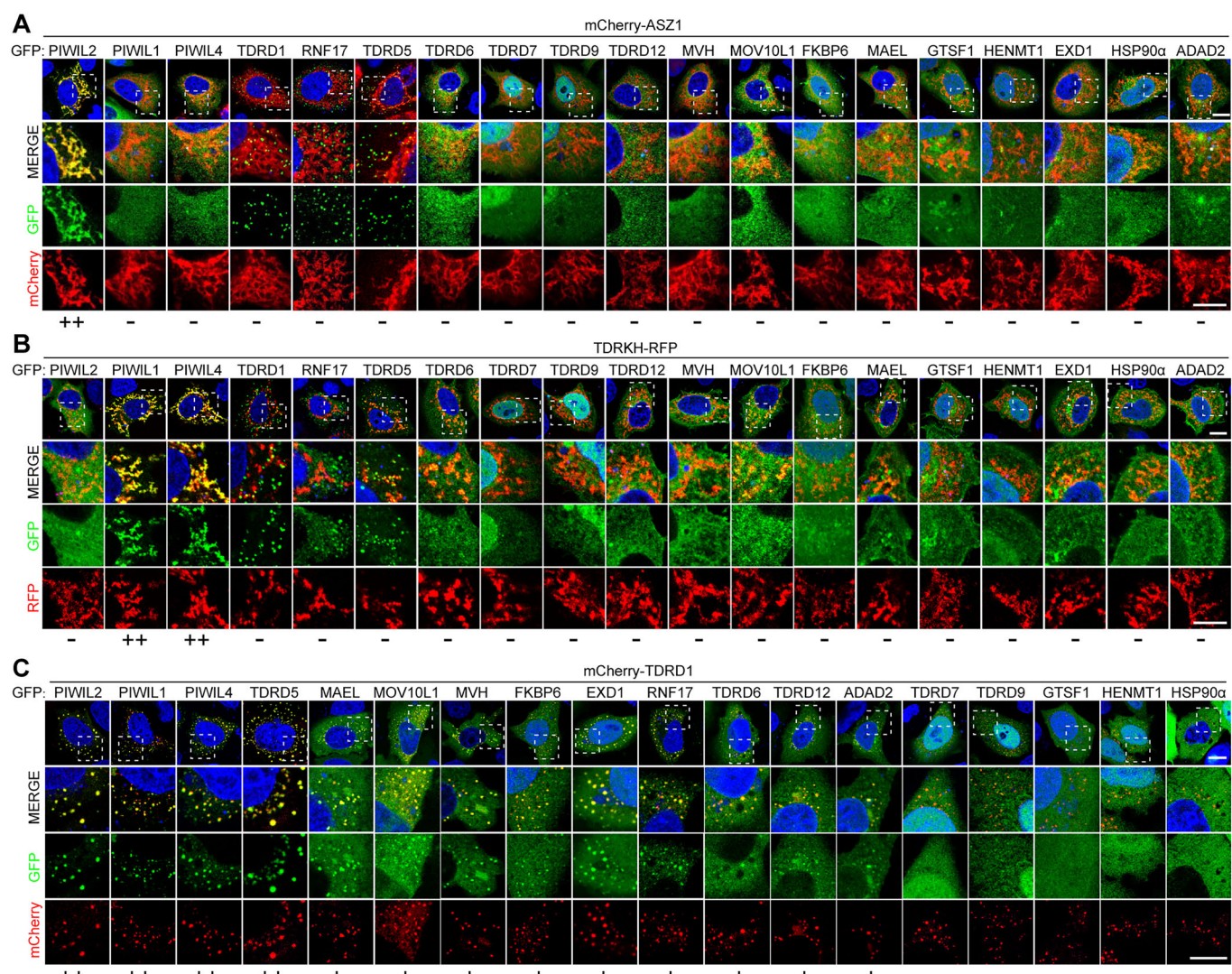

**Figure 5. Mitochondrial-anchored proteins preferentially recognize PIWI proteins in piRNA biogenesis machinery.**

(A–C) Images of HeLa cells transfected with the indicated plasmids. Dotted boxes indicate the zoomed-in areas shown below. (++, complete colocalization; +, partial colocalization; −, no or minimal colocalization.) Scale bars, 10 μm. Source data are available online for this figure.

PIWIL1 and PIWIL2 granules were diminished in *Tdrd1cKO* spermatocytes (Appendix Fig. S2G,H). In early (stage I–VI) and middle (stage VII–VIII) pachytene spermatocytes, the majority of PIWIL2 colocalized with TDRKH in both control and *Tdrd1cKO* mice (Fig. 6D). In late pachytene or diplotene spermatocytes (stage IX–XI), the dissociation of PIWIL2 from mitochondria was more pronounced in *Tdrd1cKO* mice (Fig. 6D). We observed similar results by PIWIL1 and TDRKH co-immunostaining (Fig. 6E). These data indicate that loss of TDRD1 does not affect the initial recruitment of PIWIL2 and PIWIL1 but destabilizes the piRNA biogenesis machinery and promotes piRNA complex dissociation from mitochondria in spermatocytes. TEM analysis revealed that the

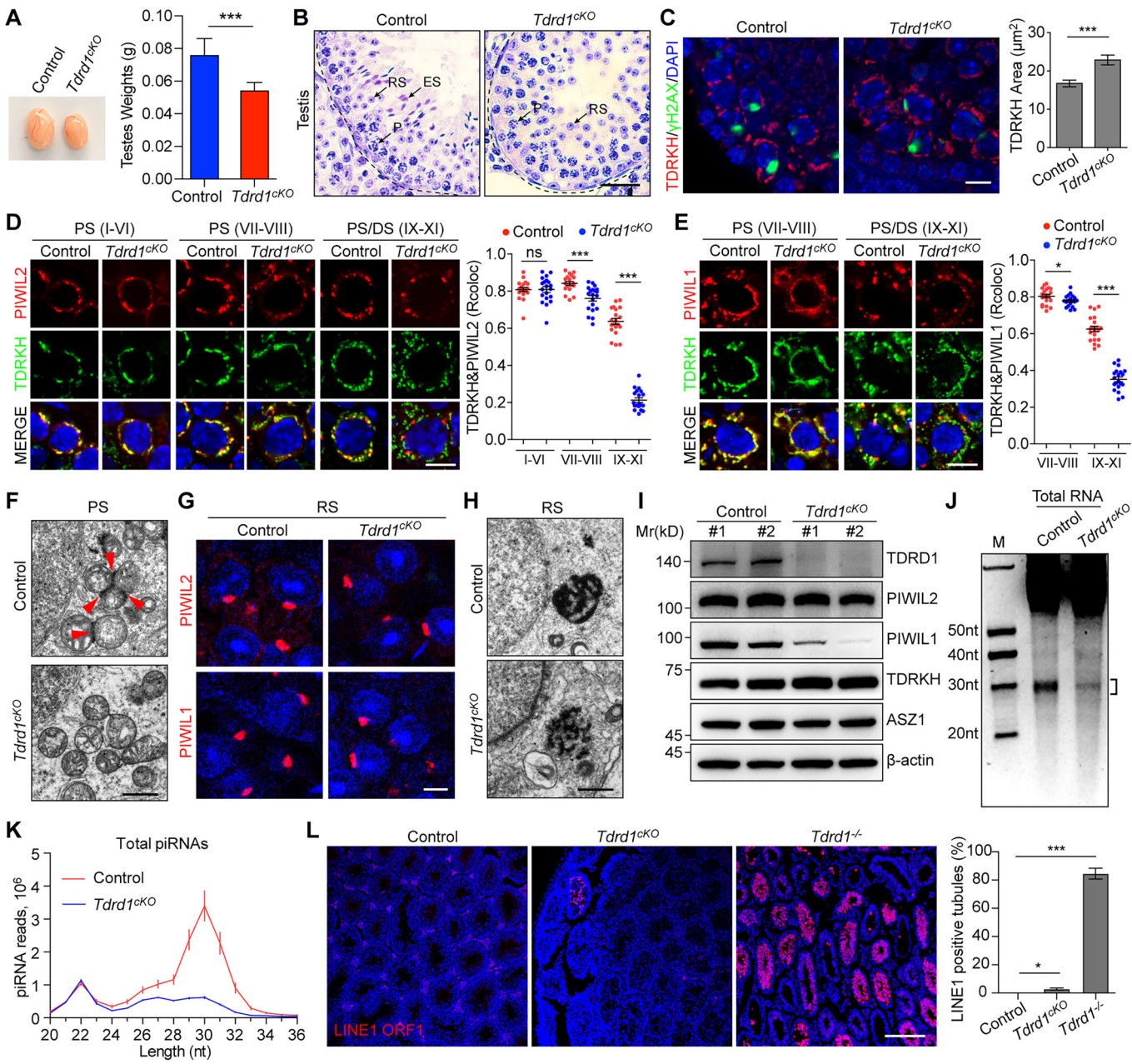

**Figure 6. TDRD1 plays irreplaceable roles in IMC assembly and pachytene piRNA biogenesis in adult testes.**

(A) Left, a representative image of testes from indicated mice; right, the average weight of adult testes ($n = 6$; ***$p < 0.001$). (B) H&E staining on adult testes. P pachytene, RS round spermatid, ES elongated spermatid. Scale bars, 50 μm. (C) Co-immunostaining of TDRKH and γH2AX on adult testes. Scale bars, 10 μm. Mitochondrial areas corresponding to TDRKH immunostaining signal in each pachytene spermatocyte (VII–VIII) are shown on the right. ($n = 20$; ***$p < 0.001$). (D, E) Co-immunostaining of PIWIL2-TDRKH (D) or PIWIL1-TDRKH (E) on adult testes. PS pachytene spermatocytes, DS diplotene spermatocytes. Scale bars, 10 μm. Quantification of the colocalization ratio is shown on the right. ($n = 50$; *$p < 0.05$; ***$p < 0.001$; ns not significant). The developmental stages of germ cells were distinguished according to DAPI staining. (F) Images of TEM on pachytene spermatocytes (PS) from adult testes. IMCs were indicated by red arrowheads. Scale bars, 1 μm. (G) Immunostaining of PIWIL2 and PIWIL1 on round spermatids (RS) from adult testes. Scale bars, 5 μm. (H) Images of TEM on round spermatids (RS) from adult testes to show CB structure. Scale bars, 1 μm. (I) WB of TDRD1, PIWIL2, PIWIL1, TDRKH, and ASZ1 in adult testes. β-actin served as a control. (J) Total RNAs from adult testes were detected by urea-PAGE gel. Square bracket indicates the piRNA population. (K) The length distribution of small RNAs from adult testes. Data were normalized by miRNA reads (21–23 nt). $n = 3$. (L) Immunostaining of LINE1 ORF1 on adult testes. Quantification of the LINE1-positive tubules per testis was shown on the right ($n = 3$; *$p < 0.05$; ***$p < 0.001$). Scale bars, 200 μm. Data information: In (A, C, D, E, K, L), data were presented as mean ± s.e.m. In (A, C, D, E, L), p values were calculated using Student's t-test (*$p < 0.05$; ***$p < 0.001$; ns not significant). $p = 5.797E-07$ (A); $p = 0.0004$ (C); p values from left to right (D): $p = 0.9853$, $p = 6.9797E-05$, $p = 1.8520E-22$; p values from left to right (E): $p = 0.0429$, $p = 3.9257E-16$; p values from left to right (L): $p = 0.0227$, $p = 2.5202E-05$. Source data are available online for this figure.

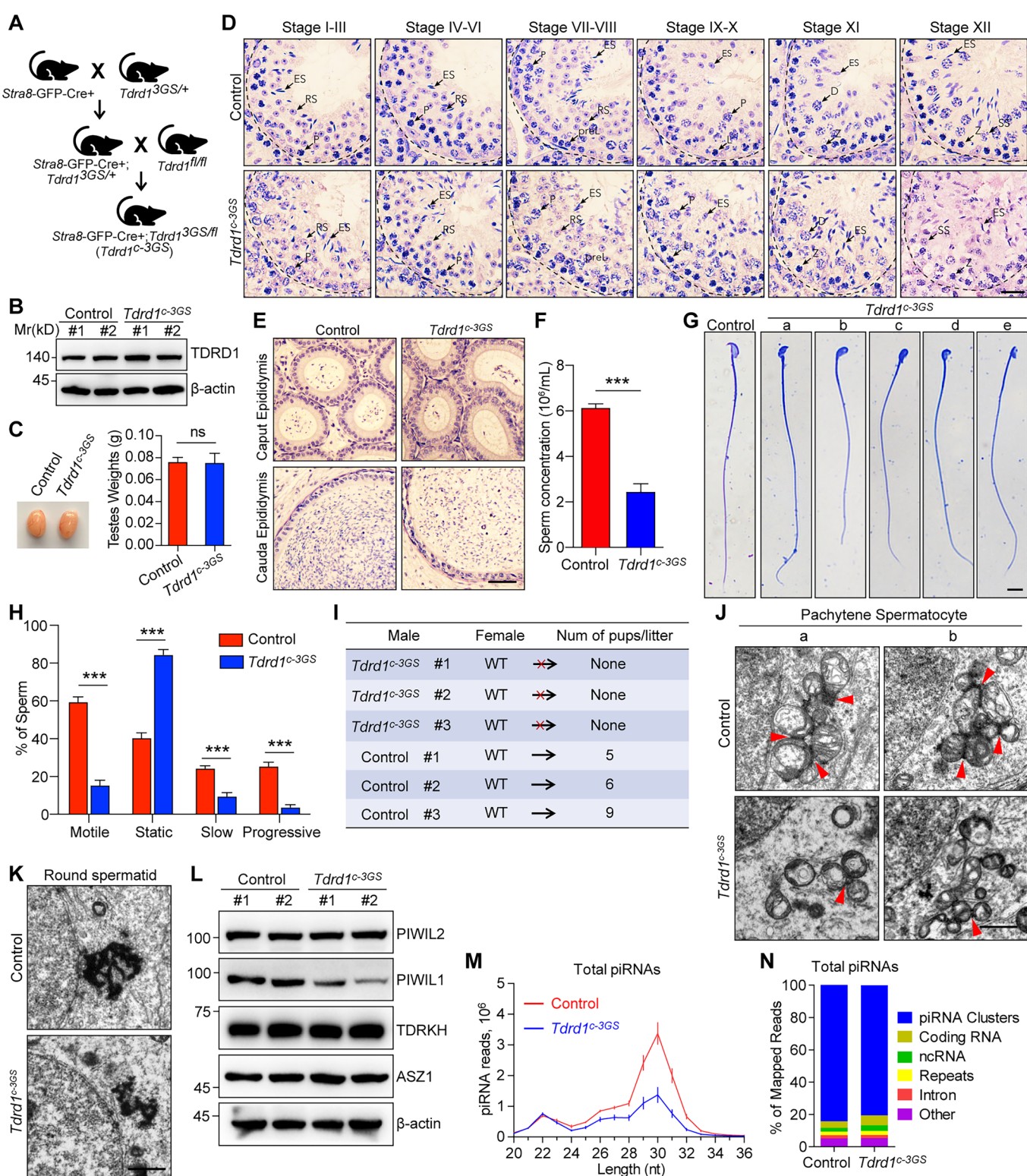

IMC structure was completely absent in *Tdrd1^cKO* pachytene spermatocytes (Fig. 6F), highlighting the pivotal role of TDRD1 in IMC assembly. Immunostaining further demonstrated that the localization of PIWIL1 or PIWIL2 in CBs was not affected in *Tdrd1^cKO* round spermatids, suggesting that TDRD1 is not required for the initiation of CB assembly (Fig. 6G). However, CBs were fragmentated in *Tdrd1^cKO* round spermatids by TEM, indicating that TDRD1 is critical for CB integrity (Fig. 6H). Together, we conclude that TDRD1 plays a crucial role in IMC assembly in pachytene spermatocytes.

**Figure 7.    TDRD1 phase separation is required for IMC assembly, pachytene piRNA biogenesis, and spermiogenesis in adult testes.**

(A) The breeding scheme to generate *Tdrd1^{c-3GS}* mice. (B) WB of TDRD1 expression in control and *Tdrd1^{c-3GS}* adult testes. β-actin served as a control. (C) Left, a representative image of testes from indicated mice; right, the average weight of adult testes. (*n* = 6; ns not significant). (D) H&E staining on adult testes. preL preleptotene, L leptotene, Z zygotene, P pachytene, D diplotene, RS round spermatid, ES elongated spermatid, SS secondary spermatocytes. Scale bars, 20 μm. (E) H&E staining on the epididymis from adult mice. Scale bars, 50 μm. (F) Analysis of sperm concentration from the adult mouse epididymis. (*n* = 9; ***p < 0.001). (G) Coomassie Blue G-250 staining of sperms from adult mouse epididymis. Scale bars, 10 μm. (H) Analysis of sperm motility from adult mouse epididymis. (*n* = 9; ***p < 0.001). (I) Fertility testes for control and *Tdrd1^{c-3GS}* adult male mice. (J, K) Images of TEM on pachytene spermatocytes (J) and round spermatids (K). IMCs were indicated by red arrowheads. Scale bars, 1 μm. (L) WB of PIWIL2, PIWIL1, TDRKH, and ASZ1 in adult testes. β-actin served as a control. (M) The length distribution of small RNAs from adult testes. Data were normalized by miRNA reads (21–23 nt). *n* = 3. (N) Genomic annotation of piRNAs from control and *Tdrd1^{c-3GS}* adult testes. Data were representative of three biological replicates. Data information: In (C, F, H, M), data were presented as mean ± s.e.m. In (C, F, H), *p* values were calculated using Student's *t*-test (***p < 0.001; ns not significant). *p* = 0.9334 (C); *p* = 2.3583E-07 (F); *p* values from left to right (H): *p* = 2.5589E-09, *p* = 2.5589E-09, *p* = 6.0999E-06, *p* = 1.6962E-07. Source data are available online for this figure.

We next studied the pachytene piRNA processing deficiency in *Tdrd1^{cKO}* mice. WB results showed that PIWIL1 protein level were dramatically decreased in *Tdrd1^{cKO}* testes, while PIWIL2, TDRKH, and ASZ1 were unaffected (Fig. 6I). We isolated total RNA from testes and used Urea-PAGE to detect piRNAs. The results showed a significant reduction in the abundance of total piRNA in *Tdrd1^{cKO}* testes (Fig. 6J). We performed small RNA sequencing and found that the piRNA reads (24–32 nt) were significantly diminished in *Tdrd1^{cKO}* testes (Fig. 6K). Mapping the piRNA reads to the mouse genome revealed that the majority of pachytene piRNAs are derived from piRNA clusters in both control and *Tdrd1^{cKO}* mice (Appendix Fig. S2I). Together, we demonstrate that TDRD1 plays critical roles in pachytene piRNA biogenesis.

TDRD1 promotes fetal piRNA biogenesis to repress transposon (Chuma et al, 2006; Reuter et al, 2009). We next assessed the LINE1 expression in *Tdrd1^{cKO}* testes. Immunostaining for LINE1 ORF1 revealed that the majority of germ cells did not exhibit LINE1 expression, only a few seminiferous tubules showed LINE1 positivity in *Tdrd1^{cKO}* adult testes. As a control, most of the seminiferous tubules were LINE1 positive in *Tdrd1^{−/−}* testes (Fig. 6L). WB results further confirmed that LINE1 had minimal upregulation in *Tdrd1^{cKO}* testes (Appendix Fig. S2J). These data indicate that TDRD1 is not required for the majority of germ cells to silence LINE1 in adult testes.

## TDRD1 phase separation is required for IMC assembly, pachytene piRNA biogenesis, and spermiogenesis in adult testes

TDRD1 phase separation plays essential roles in fetal/neonatal gonocytes (Gao et al, 2024). To investigate the physiological importance of TDRD1 phase separation in adult testes, we combined the *Tdrd1* conditional allele (*Tdrd1^{fl}*) and *Tdrd1* phase separation-deficient allele (*Tdrd1^{3GS}*) with *Stra8*-GFP-Cre to generate *Tdrd1* conditional phase separation inactivation mice (*Stra8*-GFP-Cre; *Tdrd1^{fl/3GS}*, referred to as *Tdrd1^{c-3GS}*) (Fig. 7A). WB and immunostaining using TDRD1 antibody revealed that TDRD1-3GS mutation did not alter the TDRD1 stability in adult testes (Figs. 7B and EV5A). *Tdrd1^{c-3GS}* adult mice exhibited comparable testis weight compared to control mice (Fig. 7C). H&E staining revealed that the spermatogenesis appeared to be normal in *Tdrd1^{c-3GS}* adult testes as we observed all types of germ cells (Fig. 7D). By performing ACRV1 immunostaining, the round spermatids (step1–8) exhibited no defects in *Tdrd1^{c-3GS}* testes. Thereafter, the development of elongated spermatids appeared to be delayed (Fig. EV5B). In contrast with *Tdrd1^{cKO}*, the apoptosis was

not increased in *Tdrd1^{c-3GS}* testes (Fig. EV5C). However, the sperm counts were significantly decreased in *Tdrd1^{c-3GS}* epididymides (Fig. 7E,F). Morphological analysis exhibited the abnormal heads on mature sperms in *Tdrd1^{c-3GS}* epididymides (Fig. 7G). Consistent with the abnormal morphology, the motility of sperms was severely impaired in *Tdrd1^{c-3GS}* mice (Fig. 7H). As a result, *Tdrd1^{c-3GS}* male mice showed complete infertility (Fig. 7I). These data demonstrate that TDRD1 phase separation is required for spermiogenesis in mice.

We further investigated the potential role of TDRD1 phase separation in IMC and CB formation in adult testes. Co-staining analysis revealed that TDRD1 perfectly colocalized with PIWIL2 in both control and *Tdrd1^{c-3GS}* pachytene spermatocytes (Fig. EV5D). However, in *Tdrd1^{c-3GS}* round spermatids, the TDRD1-3GS mutant displayed a dispersed cytoplasmic distribution rather than CB localization (Fig. EV5E). This indicates that TDRD1 phase separation is not required for its initial mitochondrial recruitment but is important for its CB enrichment. In late pachytene or diplotene spermatocytes (stage IX–XI), the dissociation of PIWIL1 and PIWIL2 from mitochondria was more pronounced in *Tdrd1^{c-3GS}* mice (Fig. EV5F,G), which is reminiscent of the results in *Tdrd1^{cKO}* mice. By performing PIWIL1 and PIWIL2 IP in WT and *Tdrd1^{c-3GS}* testes, TDRD1-3GS mutant showed significantly reduced interaction with PIWIL1 and PIWIL2 in testes (Fig. EV5H). TEM analysis revealed that both the IMC in pachytene spermatocytes and CB in round spermatids exhibited impaired integrity in *Tdrd1^{c-3GS}* mice, respectively (Fig. 7J,K). Collectively, these data demonstrate that TDRD1 phase separation is required for both IMC and CB integrity in adult testes.

The defects of IMC and CB implied the deficiency of pachytene piRNA biogenesis. Indeed, WB revealed that PIWIL1 protein level was decreased in *Tdrd1^{c-3GS}* testes (Fig. 7L). By small RNA sequencing, the piRNA population was diminished in *Tdrd1^{c-3GS}* testes (Fig. 7M). The source of piRNA was unaffected in *Tdrd1^{c-3GS}* mice (Fig. 7N). Immunostaining and WB revealed that LINE1 expression was not up-regulated in *Tdrd1^{c-3GS}* mice (Fig. EV5I,J). Together, these data indicate that TDRD1 phase separation is required for pachytene piRNA biogenesis.

## Discussion

To date, dozens of proteins have been implicated in piRNA biogenesis and/or functioning. While the molecular functions of individual piRNA pathway proteins are progressively being

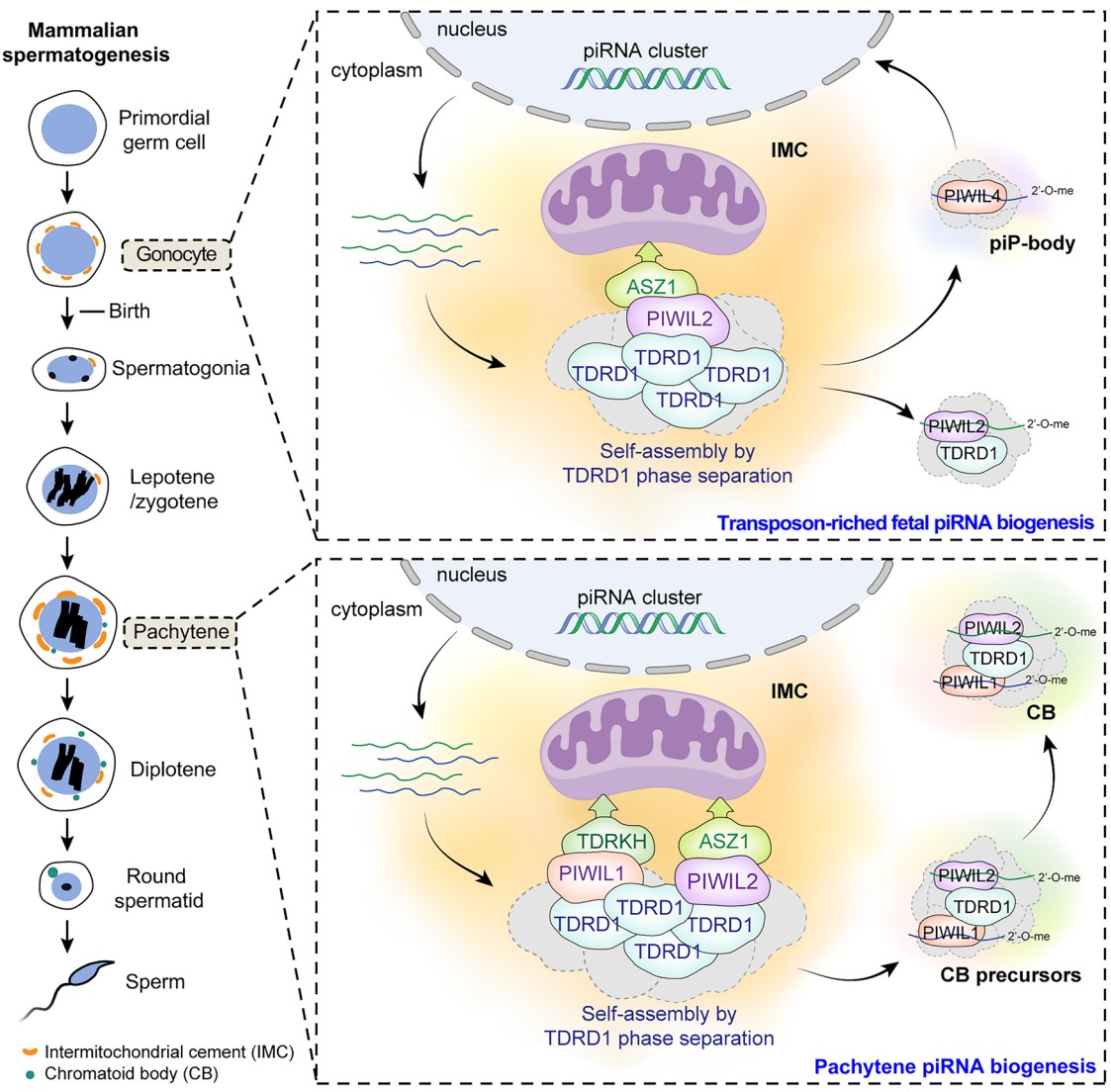

**Figure 8. Schematic model showing the role of ASZ1-PIWIL2-TDRD1 complex and TDRKH-PIWIL1-TDRD1 complex in IMC assembly during mammalian spermatogenesis.**

In fetal gonocytes, ASZ1 specifically recognizes and recruits PIWIL2 to mitochondria. PIWIL2 simultaneously interacts with ASZ1 and TDRD1 to form the ASZ1-PIWIL2-TDRD1 complex. In pachytene spermatocytes, PIWIL1 and PIWIL2 are anchored to mitochondria via interactions with TDRKH and ASZ1, respectively, and together they recruit TDRD1 to IMCs. Subsequently, TDRD1 undergoes phase separation to promote IMC assembly among clustered mitochondria. After piRNA processing, piRNA loading onto PIWI proteins (PIWIL1 and PIWIL2) results in the dissociation of PIWI proteins from mitochondrial-anchored proteins.

elucidated, our understanding of the organization and coordination of piRNA biogenesis machinery remains limited. In this study, we systematically analyzed the protein interactions of core IMC components and identified the seed complex that triggers the formation of the piRNA biogenesis complex during spermatogenesis. We demonstrated that PIWI proteins act as bridges, tethering non-mitochondrial proteins to IMCs. In fetal gonocytes, the mitochondrial-anchored protein ASZ1 specifically recognizes and recruits PIWIL2 to mitochondria. PIWIL2 simultaneously interacts with ASZ1 and TDRD1 to form the ASZ1-PIWIL2-TDRD1 complex. In pachytene spermatocytes, PIWIL1 and PIWIL2 are anchored to mitochondria via interactions with TDRKH and ASZ1, respectively, and together they recruit TDRD1 to IMCs. The other

non-mitochondrial piRNA processing proteins are expected to be incorporated into IMCs through interactions with PIWIL2-TDRD1 and/or PIWIL1-TDRD1 complexes. Subsequently, TDRD1 undergoes phase separation to promote IMC assembly among clustered mitochondria. After piRNA processing, piRNA loading onto PIWI proteins (PIWIL1 and PIWIL2) results in the dissociation of PIWI proteins from mitochondrial-anchored proteins, ultimately leading to IMC disassembly (Wei et al, 2024). Collectively, our results reveal the mechanism underlying the dynamics of piRNA biogenesis complex formation (Fig. 8).

IMC and CB are the two most prominent germ granules during mammalian spermatogenesis, serving as the platforms for piRNA biogenesis and piRNA functioning, respectively (Wei et al, 2024).

Notably, the assembly mechanisms of IMC and CB are strikingly different. IMC formation is the prerequisite for piRNA biogenesis, and its abnormal disassembly causes piRNA biogenesis deficiency. By contrast, piRNA processing is dispensable for IMC assembly and integrity, as evidenced by the normal IMC structure observed in several piRNA biogenesis-deficient mice (Ding et al, 2019; Wei et al, 2023). CBs, on the other hand, do not appear to be responsible for piRNA biogenesis, as they lack several key piRNA processing proteins. PIWI-piRNA complex are generated in IMCs and subsequently translocate into CBs for piRNA targeting and functioning (Wei et al, 2024). Interestingly, while piRNA abundance is not required for CB initiation, it is essential for maintaining CB integrity. We therefore propose that free PIWI proteins are predominantly enriched in IMCs, while PIWI-piRNA complexes are specifically enriched in CBs. The precise role of PIWl-piRNA complexes in CB formation requires further investigation.

One of the most striking findings in our work is the specificity of the interaction between PIWI proteins and mitochondrial-anchored proteins. In addition to PIWI proteins, other non-mitochondrial proteins do not appear to strongly interact with the known mitochondrial-anchored IMC components. PIWI proteins may act as bridges, tethering other components to IMCs. We hypothesize that this interaction specificity governs the dynamics of the entire piRNA biogenesis machinery in IMCs. The assembly of the piRNA biogenesis machinery is orchestrated by the recruitment of free PIWI proteins via ASZ1 and TDRKH. The piRNA loading leads to dissociation of PIWI-piRNA complexes from IMCs, resulting in the release of IMC components, which then co-translocate with PIWI-piRNA complexes to CBs. This is supported by the observation that many IMC components relocate to CBs in association with PIWI-piRNA complexes (Meikar et al, 2014; Olotu et al, 2023). However, not all components of the IMCs are capable of translocating to CBs. For instance, MOV10L1, TDRD12, and FKBP6 are exclusively localized in IMCs and are absent in CBs (Olotu et al, 2023). Interestingly, our co-IP experiments revealed a weak interaction between MOV10L1 and ASZ1 (Fig. EV1B). We cannot exclude the possibility that mitochondrial-anchored proteins may partially recruit IMC components through weak interactions in mouse germ cells.

In fetal gonocytes, piRNAs are crucial for transposon LINE1 silencing. The PIWIL4-piRNA complex promotes LINE1 DNA methylation in the nucleus to mediate transcriptional repression of LINE1, while the PIWIL2-piRNA complex degrades LINE1 mRNA through a post-transcriptional mechanism (Ernst et al, 2017). The role of pachytene piRNAs in LINE1 silencing is still controversial. Both PIWIL2 and PIWIL1 are reported to be required for LINE1 silencing through piRNA-guided slicer activity in adult testes (De Fazio et al, 2011; Reuter et al, 2011). However, the majority of germ cells are able to efficiently repress LINE1 in several pachytene piRNA biogenesis-deficient mice (Ding et al, 2019; Wei et al, 2023; Zheng and Wang, 2012). This suggests that the abundance of pachytene piRNAs may not be as critical as previously thought, and even a substantial reduction in their levels may still be sufficient to silence LINE1 in adult testes. One potential explanation is that PIWIL1 and PIWIL2 cooperate and provide redundancy in repressing LINE1, thereby ensuring genome stability during meiosis. The precise regulatory role of pachytene piRNA in LINE1 silencing requires further investigation.

# Methods

### Reagents and tools table

| Reagent/resource | Reference or source | Identifier or catalog number |
|---|---|---|
| **Experimental models** | | |
| Piwil2$^{-/-}$ mice | (Gao et al, 2024) | N/A |
| Piwil2$^{cKO}$ mice | This paper | N/A |
| Piwil1$^{-/-}$ mice | This paper | N/A |
| Tdrd1$^{cKO}$ mice | This paper | N/A |
| Tdrd1$^{-/-}$ mice | (Gao et al, 2024) | N/A |
| Tdrd1$^{c-3GS}$ mice | This paper | N/A |
| Human: 293T | TCCCAS | SCSP-502 |
| Human: HeLa | TCCCAS | SCSP-504 |
| Escherichia coli DH5α chemically competent cell | Tsingke | Cat# TSC-C01 |
| **Recombinant DNA** | | |
| See Appendix Table S1 | | |
| **Antibodies** | | |
| Rabbit polyclonal anti-PIWIL2 | MBL | Cat# PM044, RRID: AB_1279201 |
| Rabbit polyclonal anti-PIWIL1 | Cell Signaling Technology | Cat# 2079S, RRID: AB_2165432 |
| Rabbit polyclonal anti-TDRKH | Proteintech | Cat# 13528-1-AP, RRID: AB_2303299 |
| Rabbit polyclonal anti-ASZ1 | Proteintech | Cat# 21550-1-AP, RRID: AB_10858622 |
| Mouse polyclonal anti-PIWIL2 | Santa Cruz | Cat# sc-377258 |
| Rabbit polyclonal anti-TDRD1 | (Gao et al, 2024) | N/A |
| Rabbit polyclonal anti-LINE1 ORF1 | (Ding et al, 2017) | N/A |
| Mouse FITC-conjugated anti-γH2AX | Millipore | Cat# 16-202 A, RRID: AB_568825 |
| Rabbit polyclonal anti-ACRV1 | Proteintech | Cat# 14040-1-AP, RRID: AB_10640426 |
| Rabbit polyclonal anti-SYM10 | Millipore | Cat# 07-412, RRID: AB_310594 |
| Mouse polyclonal anti-SYM24 | Millipore | Cat# 07-414, RRID: AB_310596 |
| HRP-conjugated mouse monoclonal anti-Flag | ABclonal | Cat# AE024, RRID: AB_2769864 |
| Rabbit monoclonal anti-GFP | ABclonal | Cat# AE078 |
| Mouse monoclonal anti-HA | ABclonal | Cat# AE008, RRID: AB_2770404 |
| Mouse monoclonal anti-Flag | ABclonal | Cat# AE005, RRID: AB_2770401 |
| Mouse monoclonal anti-β-actin | ABclonal | Cat# AC004, RRID:AB_2737399 |
| Rabbit control IgG | ABclonal | Cat# AC005, RRID:AB_2771930 |

| Reagent/resource | Reference or source | Identifier or catalog number |
|---|---|---|
| **Chemicals, enzymes, and other reagents** | | |
| RiboLock RNase Inhibitor | Thermo Fisher | Cat# EO0382 |
| Proteinase inhibitor cocktail | Thermo Fisher | Cat# A32963 |
| RNase A | Beyotime | Cat# ST576 |
| Protein A/G Agarose Beads | Beyotime | Cat# P2055 |
| Dimethyl sulfoxide | Beyotime | Cat# A600163-0250 |
| methylthioadenosine | Sigma | Cat# 2457-80-9 |
| TRIzol | Thermo Fisher | Cat# 15596026 |
| EZ Trans | Life-iLab | Cat# AC04L092 |
| jetPRIME | Polyplus | Cat# 101000046 |
| Hematoxylin and Eosin | Beyotime | Cat# E607318-0200 |
| mitoTracker™ Green CMXRos | Thermo Fisher | Cat# M7514 |
| mitoTracker™ Red CMXRos | Thermo Fisher | Cat# M7512 |
| Antifade mounting medium with DAPI | Beyotime | Cat# P0131 |
| **Software and algorithms** | | |
| Cutadapt v4.4 | | https://pypi.org/project/cutadapt/4.4/ |
| Bowtie v1.0.0 | | https://sourceforge.net/projects/bowtie-bio/files/bowtie/1.0.0/ |
| GraphPad Prism | GraphPad Software | https://www.graphpad.com |
| Python | | https://www.python.org/ |
| Cytoscape | (Otasek et al, 2019) | https://cytoscape.org/ |
| AlphaFold3 | (Abramson et al, 2024) | https://alphafoldserver.com/ |
| PyMOL | | https://pymol.org/ |
| ImageJ | NIH | https://imagej.nih.gov/ij/ |
| **Other** | | |
| ClonExpress Ultra One Step Cloning Kit | Vazyme | Cat# C115-01 |
| TUNEL BrightRed Apoptosis Detection Kit | Vazyme | Cat# A113-01 |
| Zenon Alexa Fluor 488 Rabbit IgG labeling Kit | Thermo Fisher | Cat# Z25302 |
| VAHTS Small RNA Library Prep Kit for Illumina V2 | Vazyme | Cat# NR811-01 |

## Cell culture and transfection

HeLa and 293T cells were obtained from the cell bank of the Type Culture Collection of the Chinese Academy of Science (TCCCAS). The cells were cultured in DMEM (Corning) supplemented with 10% fetal bovine serum, 1% penicillin-streptomycin, and maintained at 37 °C in a 5% $CO_2$ humidified incubator. Transfections were performed using EZ Trans (Life-iLab) for 293T cells and jetPRIME (Polyplus) for HeLa cells, according to the manufacturers' instructions.

## Bacterial strain

The cloning and expression of recombinant plasmids were carried out using *E. coli* DH5α chemically competent cells (Tsingke). Cells were cultured in LB medium with the appropriate antibiotics.

## Mice

*Tdrd1$^{fl/fl}$* mice, *Piwil2$^{fl/fl}$* mice, *Tdrd1$^{-/-}$* mice, *Piwil2$^{-/-}$* mice, and *Tdrd1$^{3GS}$* mutant mice were generated as previously described (Gao et al, 2024). To obtain *Tdrd1* conditional knockout mice (*Tdrd1$^{cKO}$*) or *Piwil2* conditional knockout mice (*Piwil2$^{cKO}$*), germ cell-specific *Stra8*-GFP-Cre mice (a gift from M. Tong) (Chen et al, 2018) were bred with *Tdrd1$^{fl/fl}$* mice or *Piwil2$^{fl/fl}$* mice separately. To generate *Tdrd1* conditional mutant mice (*Tdrd1$^{c-3GS}$*), *Stra8*-GFP-Cre mice were bred with *Tdrd1$^{3GS/+}$* mice to get *Stra8*-GFP-Cre+; *Tdrd1$^{3GS/+}$* mice. Male *Stra8*-GFP-Cre+; *Tdrd1$^{3GS/+}$* mice were bred with female *Tdrd1$^{fl/fl}$* mice to obtain *Stra8*-GFP-Cre+; *Tdrd1$^{3GS/fl}$* mice (*Tdrd1$^{c-3GS}$*).

To obtain *Piwil1* knockout (*Piwil1$^{-/-}$*) mice, *Piwil1$^{+/-}$* mice were generated by CRISPR/Cas9-mediated genome engineering in C57BL/6 mice by Cyagen Biosciences (Suzhou, China). In brief, the gRNAs and Cas9 were co-injected into fertilized mouse eggs to generate targeted knockout offspring. Four gRNAs (5'-GCTA-GACGGCAGAGCTCAAGTGG-3'; 5'-GCTATACTAGGAATG-TATGGAGG-3'; 5'-GCTAGTTTCTAGTGAGTACCAGG-3'; 5'-GCTGGGCATCCCTTGGCCACTGG-3') were used to delete the entire coding region of the *Piwil1* gene. F0 founder animals were identified by PCR followed by sequence analysis. Male *Piwil1$^{+/-}$* mice were bred with female *Piwil1$^{+/-}$* mice to obtain *Piwil1$^{-/-}$* mice.

The study received approval from the Institutional Animal Care and Use Committee at Tongji University. Mice were housed in the Tongji University animal facility under SPF conditions, in compliance with institutional guidelines and ethical standards. The facility staff supplied them with standard chow and water.

## Plasmid construction

The full-length cDNAs of mouse PIWIL1, PIWIL2, PIWIL4, TDRD1, TDRD5, ASZ1, TDRKH, PNLDC1, mitoPLD, GPAT2, RNF17, TDRD6, TDRD7, TDRD9, TDRD12, MOV10L1, MVH, FKBP6, MAEL, GTSF1, HENMT1, EXD1, HSP90α, and ADAD2 were amplified via PCR. The corresponding DNA fragments were subsequently inserted into expression vectors using the ClonExpress Ultra One Step Cloning Kit (Vazyme, C115-01). The eukaryotic expression vectors employed in this study included pcDNA3-Flag (Flag at N-terminal), pcDNA3-HA (HA at N-terminal), pcDNA3-RFP (TurboRFP at N-terminal), pcDNA3-mCherry (mCherry at N-terminal), pEGFP-C1 (GFP at N-terminal), pEGFP-N1 (GFP at C-terminal), and pRFP-N1 (TurboRFP at C-terminal). To generate pRFP-N1, the GFP tag in pEGFP-N1 was replaced with TurboRFP. All oligonucleotide primers for the study were synthesized by Tsingke Biotech (Beijing,

China), and all constructs were verified by Sanger sequencing. Detailed plasmid information is provided in Appendix Table S1.

## Co-immunoprecipitation (Co-IP)

293T cells were transiently transfected with the indicated plasmids using EZ Trans (Life-iLab). After transfection, cells were harvested, washed with PBS, and lysed in lysis buffer [150 mM NaCl, 50 mM Tris-HCl (pH 7.4), 1% Triton X-100, 1 mM EDTA, and protease inhibitor cocktail (Thermo Fisher)]. The lysates were incubated on ice for 30 min and then centrifuged at 12,000 rpm for 10 min at 4 °C. The supernatants were incubated overnight with Protein A/G agarose beads (Beyotime) and the respective antibodies at 4 °C. Following incubation, the beads were washed three times with cold wash buffer [300 mM NaCl, 50 mM Tris-HCl (pH 7.4), 1% Triton X-100, 1 mM EDTA, and protease inhibitor cocktail]. The beads were then boiled in SDS loading buffer at 100 °C for 5 min before being analyzed by SDS-PAGE. Western blotting was performed using the indicated antibodies.

For immunoprecipitation with RNase A treatment, adult mouse testes were homogenized in lysis buffer A [150 mM KCl, 25 mM Tris-HCl (pH 7.4), 0.5% NP-40, 5 mM EDTA, 0.5 mM DTT, and protease inhibitor cocktail (Thermo Fisher)] and incubated with 250 μg/mL RNase A for 1 h at 4 °C. To further degrade RNA, an additional 125 μg/mL RNase A and primary antibody-coupled Protein A/G agarose beads (Beyotime) were added to the precleared lysates and incubated for 5 h at 4 °C. Following incubation, the beads were washed three times with cold wash buffer [300 mM NaCl, 50 mM Tris-HCl (pH 7.4), 1% Triton X-100, 1 mM EDTA, and protease inhibitor cocktail]. The immunoprecipitates were then boiled in SDS loading buffer at 100 °C for 5 min before being analyzed by SDS-PAGE. Western blotting was performed using the specified antibodies.

## Immunoprecipitation-mass Spectrometry (IP-MS)

Testes isolated from P20 wild-type mice (two testes per replicate, three biological replicates) were lysed and homogenized in 1 mL cold lysis buffer [150 mM NaCl, 50 mM Tris-HCl (pH 7.4), 1% Triton X-100, 1 mM EDTA, protease inhibitor cocktail (Thermo Fisher), RiboLock RNase Inhibitor (Thermo Fisher)] on ice. Lysates were sonicated by a probe sonicator (Jing Xin). After 30 min of incubation on ice, cell lysates were centrifuged at 12,000 rpm for 10 min at 4 °C. The supernatants were incubated with 2 μg TDRD1, ASZ1 (21550-1-AP, Proteintech), PIWIL2 (PM044, MBL) antibodies or Rabbit IgG control antibody (AC005; ABclonal) for 2 h at 4 °C, followed by incubation with 50 μL of prewashed Protein A/G Agarose Beads (Beyotime) for another 2 h at 4 °C. After washing the beads, the bound proteins were eluted by boiling in SDS loading buffer, separated by SDS-PAGE, and visualized by Coomassie Blue staining.

The protein bands were excised and processed for in-gel trypsin digestion as described (Li et al, 2022), desalted using SOLA™ SPE (Thermo Fisher), resuspended in 80% ACN/0.1% trifluoroacetic acid (v/v), and subjected to liquid chromatography with mass spectrometry (LC-MS) analysis. Peptides separation was performed on a PepMap C-18 column (25 cm × 75 μm, 2 μm particles, 100 Å pore size; Thermo Fisher Scientific) with a linear gradient from 2 to 55% mobile phase B over 111 min, followed by an increase from 55 to 99% mobile phase B in 9 min (mobile phase A consisted of 0.1%

formic acid in water, mobile phase B consisted of 80% acetonitrile in 0.1% formic acid), the eluted peptides were then sprayed into an Orbitrap Exploris 480 mass spectrometer (Thermo Fisher) operated in data-dependent acquisition (DDA) mode. For the full MS scan, spectra were acquired in the m/z range of 350–1500 at a resolution of 60,000. The absolute AGC value was set to 3.0e6, with a maximum injection time of 25 ms. The cycle time was set to 1.5 s, and the RF lens was set to 50%. The intensity threshold was 5.0e3. For MS2 acquisition, fragmentation was performed with a normalized collision energy of 30%, a resolution of 15,000, and a maximum injection time of 22 ms. The raw data were processed using Proteome Discoverer. Peptides were searched against the mouse UniProt database (date 15.07.2024). Peptide abundances were normalized in Python 3, where missing intensity values were imputed with randomly generated data, with the maximum of the random data corresponding to the minimum of the observed data, the minimum of the random data was set to one-fourth of the minimum observed value, and the random numbers followed a normal distribution. Statistically significant enriched proteins (fold change >4, $p < 0.05$, PEP >10) are shown in Tables EV1–3. The PPI networks were generated by Cytoscape.

## Western blotting (WB)

293T cells or mouse testes were harvested and lysed in lysis buffer [150 mM NaCl, 50 mM Tris-HCl (pH 7.4), 1% Triton X-100, 1 mM EDTA, and protease inhibitor cocktail (Thermo Fisher)]. The lysates were incubated on ice for 30 min, followed by centrifugation at 12,000 rpm for 10 min at 4 °C. The supernatant was then mixed with SDS loading buffer and heated at 100 °C for 5 min prior to SDS-PAGE. Western blotting was performed using the following primary antibodies: anti-TDRD1 (1:1000; homemade), anti-LINE1 ORF1 (1:5000; homemade), anti-PIWIL2 (1:2000; MBL, PM044), anti-PIWIL1 (1:1000; Cell Signaling Technology, 2079S), anti-TDRKH (1:1000; Proteintech, 13528-1-AP), anti-ASZ1 (1:1000; Proteintech, 21550-1-AP), anti-Flag (1:3000; AE005, ABclonal), anti-Flag HRP (1:1000; AE024, ABclonal), anti-GFP (1:10,000; AF1483, Beyotime), anti-HA (1:3000; AE008, ABclonal), anti-SYM10 (1:1000; Millipore Sigma, 07-412), anti-SYM24 (1:1000; Millipore Sigma, 07-414), and anti-β-actin (1:5000; ABclonal, AC004). Blot images were captured using the Tanon-5200 Chemiluminescent Imaging System (Tanon).

## Immunostaining of mouse testes

Mouse testes were harvested and fixed overnight at 4 °C in 4% PFA in PBS, followed by paraffin embedding. Tissue sections (5-μm thick) were cut, deparaffinized, and rehydrated. Antigen retrieval was carried out by microwaving the sections in 0.01 M Tris-EDTA buffer (pH 9.0) for 2 min. After rinsing with PBS, the sections were blocked with 5% normal goat serum (NGS) for 30 min. The tissue sections were then incubated with primary antibodies, diluted in 5% NGS, at 37 °C for 1 h. The following primary antibodies were used for immunofluorescence: rabbit anti-TDRD1 (1:200; homemade), rabbit anti-LINE1 ORF1 (1:500; homemade), rabbit anti-PIWIL2 (1:200; PM044, MBL), mouse anti-PIWIL2 (1:200; sc-377258, Santa Cruz), rabbit anti-PIWIL1 (1:200; 2079S, Cell Signaling Technology), rabbit anti-ACRV1 (1:200; 14040-1-AP, Proteintech), rabbit anti-TDRKH (1:200; 13528-1-AP, Proteintech),

rabbit anti-ASZ1 (1:200; 21550-1-AP, Proteintech), and mouse FITC-conjugated anti-γH2AX (1:500; 16-202A, Millipore). Following PBS washes, the sections were incubated with fluorescently labeled secondary antibodies at room temperature for 1 h. The sections were mounted using an antifade mounting medium with DAPI (Beyotime). Fluorescence images were acquired using a Ti2-E inverted confocal microscope (Nikon, Japan) or an FV3000 inverted confocal microscope (Olympus, Japan).

For co-staining of TDRD1 with TDRKH or PIWIL1 with TDRKH, adult testis sections were first incubated with rabbit anti-TDRD1 (1:200; homemade) or rabbit anti-PIWIL1 (1:200; 2079S, Cell Signaling Technology) antibodies, followed by incubation with Alexa Fluor 555-labeled Donkey Anti-Rabbit IgG (1:500; A-31572, Thermo Fisher). The sections were then blocked again with 5% NGS in PBS and incubated with anti-TDRKH, which was labeled using the Zenon Alexa Fluor 488 Rabbit IgG Labeling Kit (Thermo Fisher) according to the manufacturer's protocol. Finally, the sections were mounted with antifade mounting medium containing DAPI.

For SCP3 and gH2AX co-staining, testes sections were incubated with mouse anti-SCP3 (1:200; sc-74569, Santa Cruz) followed by incubation with Alexa Fluor 555-labeled Donkey Anti-mouse IgG (1:500; Sigma, SAB4600060). After washing with PBS, sections were incubated with mouse FITC-conjugated anti-gH2AX (1:500; 16-202A, Millipore). The sections were mounted using an antifade mounting medium with DAPI (Beyotime).

## Immunostaining of cultured cell

HeLa cells were transfected with the indicated plasmids using jetPRIME (Polyplus) for 36 h. The transfected cells on coverslips were then fixed with 4% PFA in PBS for 10 min at room temperature. After fixation, the cells were permeabilized with 0.5% Triton X-100 in PBS for 10 min. The cells were subsequently blocked with 5% BSA in PBS for 1 h at room temperature, followed by incubation with primary antibodies: mouse anti-Flag (1:200; AE005, ABclonal) and mouse anti-HA (1:200; AE008, ABclonal), rabbit anti-PIWIL1 (1:200; 2079S, Cell Signaling Technology) at 37 °C for 1.5 h. After washing with PBS, the cells were incubated with Alexa Fluor 555-labeled Donkey Anti-Rabbit IgG (1:500; A-31572, Thermo Fisher) and/or ABflo 647-conjugated Goat Anti-Mouse IgG (1:500; AS059, ABclonal) at room temperature for 1.5 h. Finally, the cells were mounted using an antifade mounting medium containing DAPI. Fluorescence images were captured using an FV3000 inverted confocal microscope (Olympus, Japan).

## H&E staining

Mouse testicular and epididymal tissues were harvested and fixed overnight in Bouin's solution at 4 °C, followed by paraffin embedding. Sections of paraffin-embedded tissues (5-μm thick) were then deparaffinized, rehydrated, and stained with Hematoxylin and Eosin (Beyotime) following the manufacturer's protocols.

## MitoTracker staining

MitoTracker dyes were used to label mitochondria according to the manufacturer's instructions. Briefly, cells were transiently transfected with the indicated plasmids and stained with 200 nM mitoTracker for 30 min at 37 °C. The cells were fixed using 4% PFA in PBS for 10 min at room temperature and followed by immunofluorescence staining of cultured cells as described previously.

## TUNEL assay

Mouse testes were harvested and fixed overnight at 4 °C in 4% PFA in PBS, followed by paraffin embedding. Tissue sections (5-μm thick) were cut, deparaffinized, and rehydrated. TUNEL assays were performed using the TUNEL BrightRed Apoptosis Detection Kit (Vazyme), according to the manufacturer's protocol. The sections were mounted with antifade mounting medium containing DAPI (Beyotime). Fluorescence images were captured using an FV3000 inverted confocal microscope (Olympus, Japan).

## Transmission electron microscopy (TEM)

Mouse testes or HeLa cells were harvested and fixed overnight at 4 °C in 2.5% glutaraldehyde in 0.2 M phosphate buffer (pH 7.0). Following fixation, samples were post-fixed with a mixture of 1% osmium tetroxide and 2% potassium ferricyanide for 1 h at room temperature, and then dehydrated through a graded series of ethanol solutions. The samples were infiltrated with resin and embedded at 60 °C for 48 h. Sections were collected onto copper grids and stained with uranyl acetate and lead citrate. Transmission electron microscopy (TEM) images were obtained using a Tecnai Spirit electron microscope (FEI, USA) operating at 120 kV.

## Structure prediction with AlphaFold

Modeling of PIWIL2 (1-971 aa; UniProt: Q8CDG1) and ASZ1 (1–476 aa; UniProt: Q8VD46) interaction was predicted using AlphaFold3 (https://alphafoldserver.com/) (Abramson et al, 2024) with standard settings. Random seed = 1445258117; ipTM = 0.51; pTM = 0.61. Structures were visualized with PyMol.

## Immunoprecipitation of piRNAs

Mouse testes were collected and homogenized in lysis buffer [150 mM NaCl, 20 mM HEPES (pH 7.3), 2.5 mM MgCl$_2$, 0.2% NP-40, and 1 mM DTT, protease inhibitor cocktail, RiboLock RNase Inhibitor]. After sonication, the testis lysates were centrifuged at 12,000 rpm for 10 min. The supernatants were then precleared by incubating with protein A/G agarose beads (Beyotime) for 2 h. Anti-PIWIL2 (PM044, MBL) antibodies were added, along with protein A/G agarose beads (Beyotime), and incubated for 4 h. The beads were washed three times with the lysis buffer. Immunoprecipitated RNAs were extracted from the beads using TRIzol reagent (Thermo Fisher).

## Detection of piRNAs

Total RNA was extracted from mouse testes using Trizol reagent. Total RNA (1 μg) was separated on a 15% Urea-PAGE gel. The gel was then stained using Sparkgreen (SparkJade). The fluorescent signals were captured using the Tanon-5200 Imaging System (Tanon).

## Small RNA libraries and bioinformatic analyses

Small RNA libraries were constructed from both total RNA using the VAHTS Small RNA Library Prep Kit for Illumina V2 (Vazyme), following the manufacturer's protocol. Libraries with distinct barcodes were combined and sequenced on the Illumina NovaSeq 6000 system (Novogene, China).

Sequenced reads were processed using Cutadapt to remove the sequencing adapters. The trimmed reads were then filtered based on length (24–32 nt) and mapped to various sequence categories, including piRNA clusters, coding RNAs, non-coding RNAs (ncRNAs), repeats, introns, and others. Bowtie was used for alignment, allowing for one base mismatch. Repeat sequences were classified according to RepeatMasker. RNA read counts were normalized based on miRNA counts (21–23 nt) for total small RNA sequencing.

## Quantification of mitochondrial area

Mitochondrial characteristics were quantitatively analyzed using the Mitochondria Analyzer plugin in ImageJ software. The image was first processed and subjected to thresholding, with the "Weighted Mean" method applied for local thresholding. The resulting binary image was then used as input for the "2D Analysis" function, which calculates the total mitochondrial area within the designated region.

## Quantification of Western blotting analysis

Image J software was employed for the quantification of Western blotting analysis. The signal from each band was converted into intensity values. These values were subsequently normalized and utilized to calculate fold changes, providing a basis for comparing protein expression across various samples. The samples in the same panel were derived from the same experiment, and the gels/blots were processed in parallel.

## Statistics

The number ($n$) of biological replicates or mice is specified in the individual figure legends. The experiments were not randomized, and sample sizes were not predetermined using statistical methods. Statistical analyses were conducted using unpaired Independent Samples $t$-tests, with all tests being two-sided. Results are presented as the mean ± s.e.m. $p$ values are indicated in the figures, with significance determined as follows: $*p < 0.05$, $**p < 0.01$, and $***p < 0.001$.

# Data availability

All data presented are available in the main text and supplementary materials. Data files of small RNA-seq have been uploaded to the SRA of NCBI with an accession number PRJNA1219779.

The source data of this paper are collected in the following database record: biostudies:S-SCDT-10_1038-S44318-025-00579-x.

# Peer review information

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

## Acknowledgements

We thank Prof. Minghan Tong from the Chinese Academy of Sciences for sharing the *Stra8*-GFP-Cre mice. We thank members of the D.D. lab for technical assistance and helpful comments, and the animal facility of Tongji University for the care of mice. This work was supported by grants from the National Natural Science Foundation of China (32470901) and the National Key R&D Program of China (2022YFA1305300). Chen Chen was supported by NIH grants (R01HD084494 and R35GM156209).

## Author contributions

**Jie Gao**: Conceptualization; Resources; Data curation; Formal analysis; Validation; Investigation; Methodology; Writing—original draft. **Canmei Chen**: Validation; Investigation; Methodology; Writing—review and editing. **Guanyi Shang**: Data curation; Formal analysis; Methodology. **Wenyang Yu**: Software; Methodology. **Ting Zhao**: Investigation. **Yunfang Zhang**: Data curation; Writing—review and editing. **Chen Chen**: Data curation; Funding acquisition; Writing—review and editing. **Deqiang Ding**: Conceptualization; Data curation; Formal analysis; Supervision; Funding acquisition; Investigation; Writing—original draft; Project administration; Writing—review and editing.

Source data underlying figure panels in this paper may have individual authorship assigned. Where available, figure panel/source data authorship is

listed in the following database record: biostudies:S-SCDT-10_1038-S44318-025-00579-x.

## Disclosure and competing interests statement

The authors declare no competing interests.

# Expanded View Figures

**Figure EV1.  PIWIL2 serves as an adapter to form the ASZ1-PIWIL2-TDRD1 complex.**

(**A**) Interaction networks of ASZ1 and PIWIL2 based on IP/MS from P20 testes. The known piRNA-related proteins are highlighted by red circles. (**B**) Co-IP assay in 293T cells transfected with indicated constructs. Flag-tagged and GFP-tagged proteins were detected by WB. (**C**) 293T cells were treated with methyltransferase inhibitor MTA. WB was performed using sDMA (SYM10) or aDMA (ASYM24) antibodies. β-actin served as a control. (**D, E**) Co-IP assay in transfected 293T cells with MTA treatment. Flag-tagged and HA-tagged proteins were detected by WB. (**F**) HeLa cells were transfected with indicated constructs. Immunostaining was performed using the HA antibody. Scale bars, 10 μm. Fluorescence intensity through positions denoted by the white lines are shown in right. (**G**) Image of TEM on HeLa cells transfected with GFP-TDRD1, HA-PIWIL2, or RFP-ASZ1, respectively. Scale bars, 1 μm.

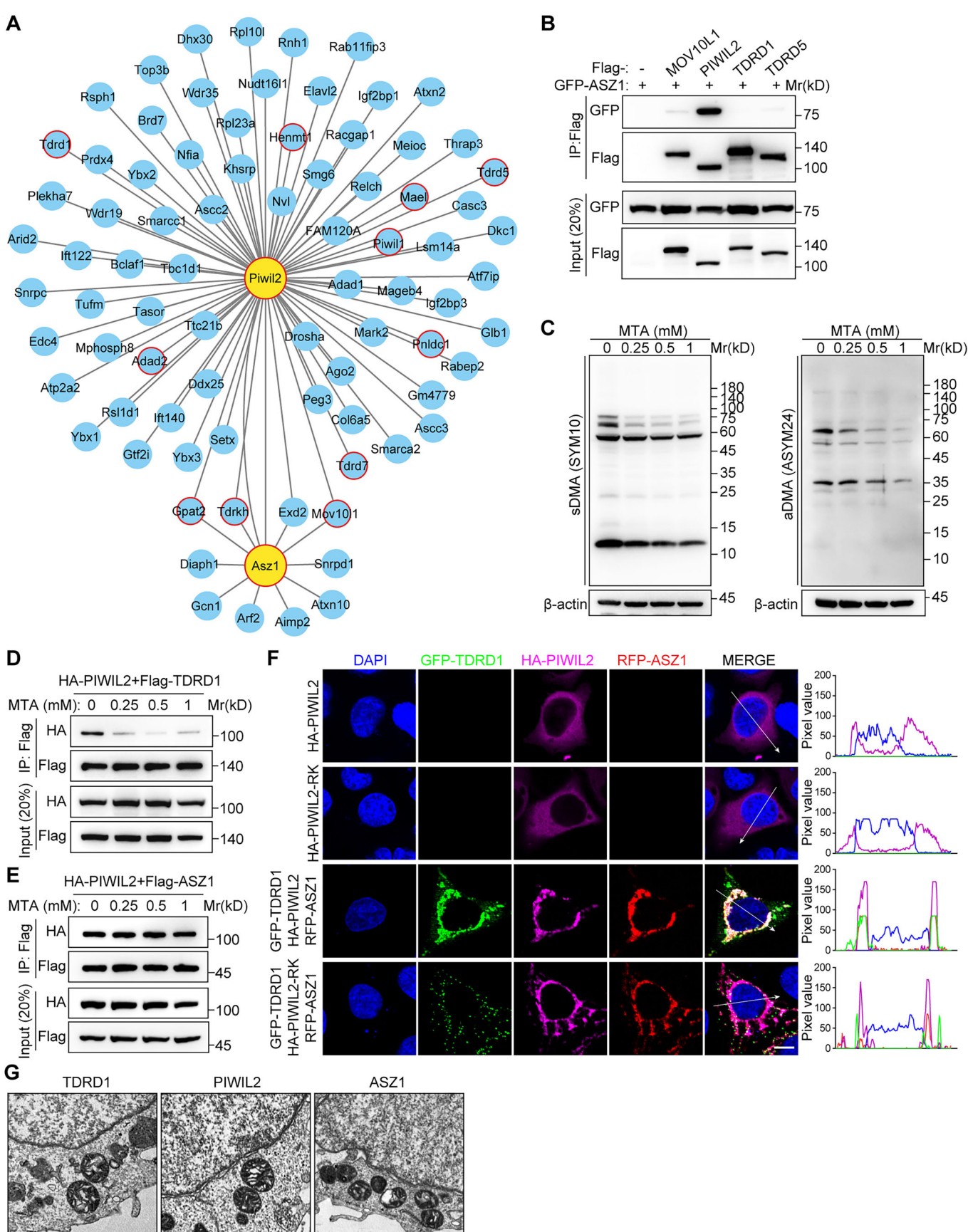

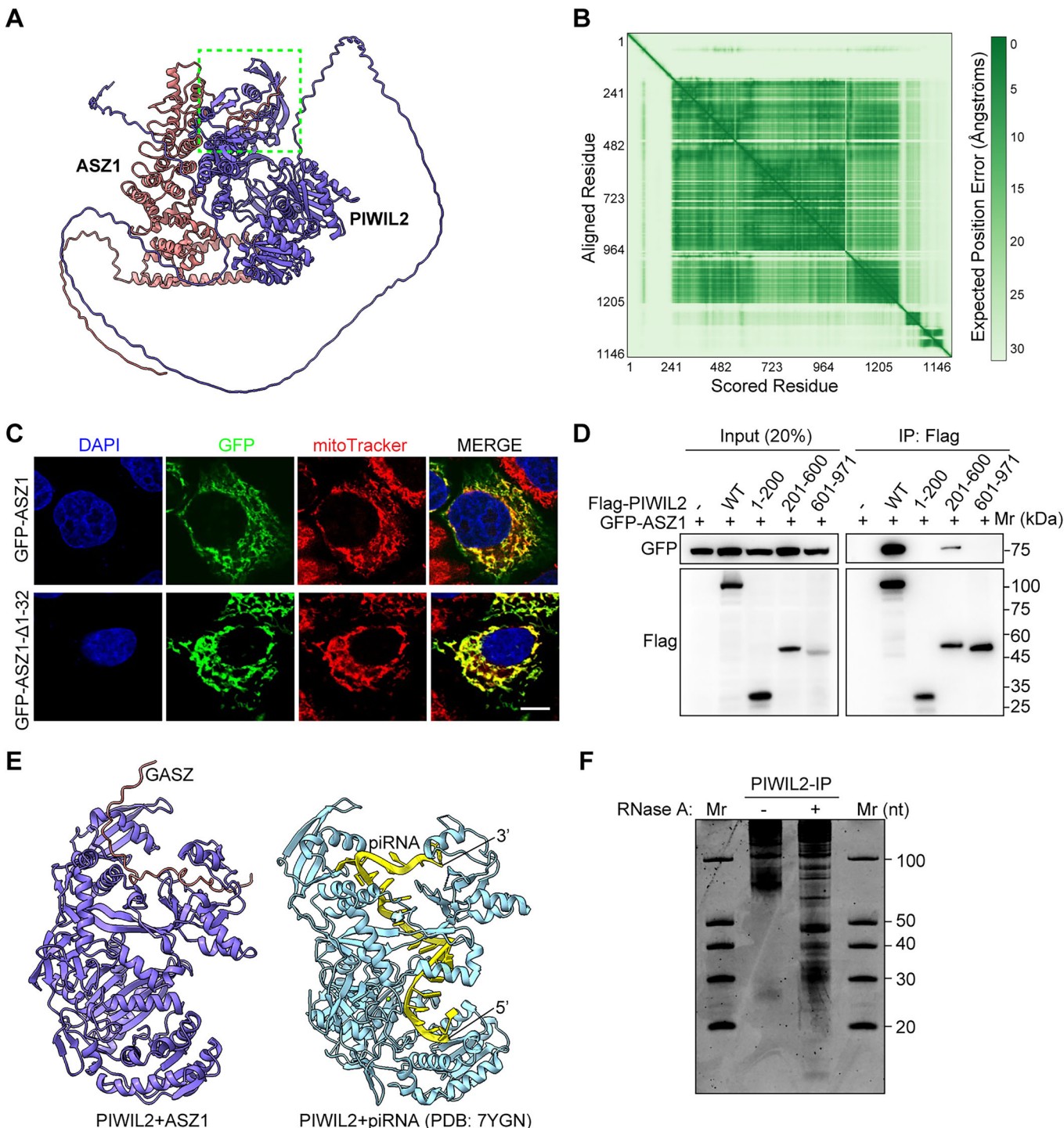

**Figure EV2.  piRNA loading onto PIWIL2 disrupts ASZ1-PIWIL2 interaction.**

(**A**) Interaction structure model of the PIWIL2-ASZ1 complex predicted by AlphaFold. The green box highlights the interaction interface. (**B**) PAE (predicted aligned error) plot showing regions of high confidence (dark green) and low confidence (pale green) for the predicted structure of PIWIL2 and ASZ1 complex. (**C**) HeLa cells were transfected with indicated constructs. Transfected cells were stained with Mitotracker to label mitochondria. Scale bars, 10 μm. (**D**) Co-IP assay in 293T cells transfected with indicated constructs. Flag-tagged and GFP-tagged proteins were detected by WB. (**E**) Left, PIWIL2-ASZ1 structure predicted by AlphaFold; right, PIWIL2-piRNA structure from PDB (7YGN). (**F**) PIWIL2 bound RNAs from adult testes with or without RNase A treatment were detected by Urea-PAGE gel.

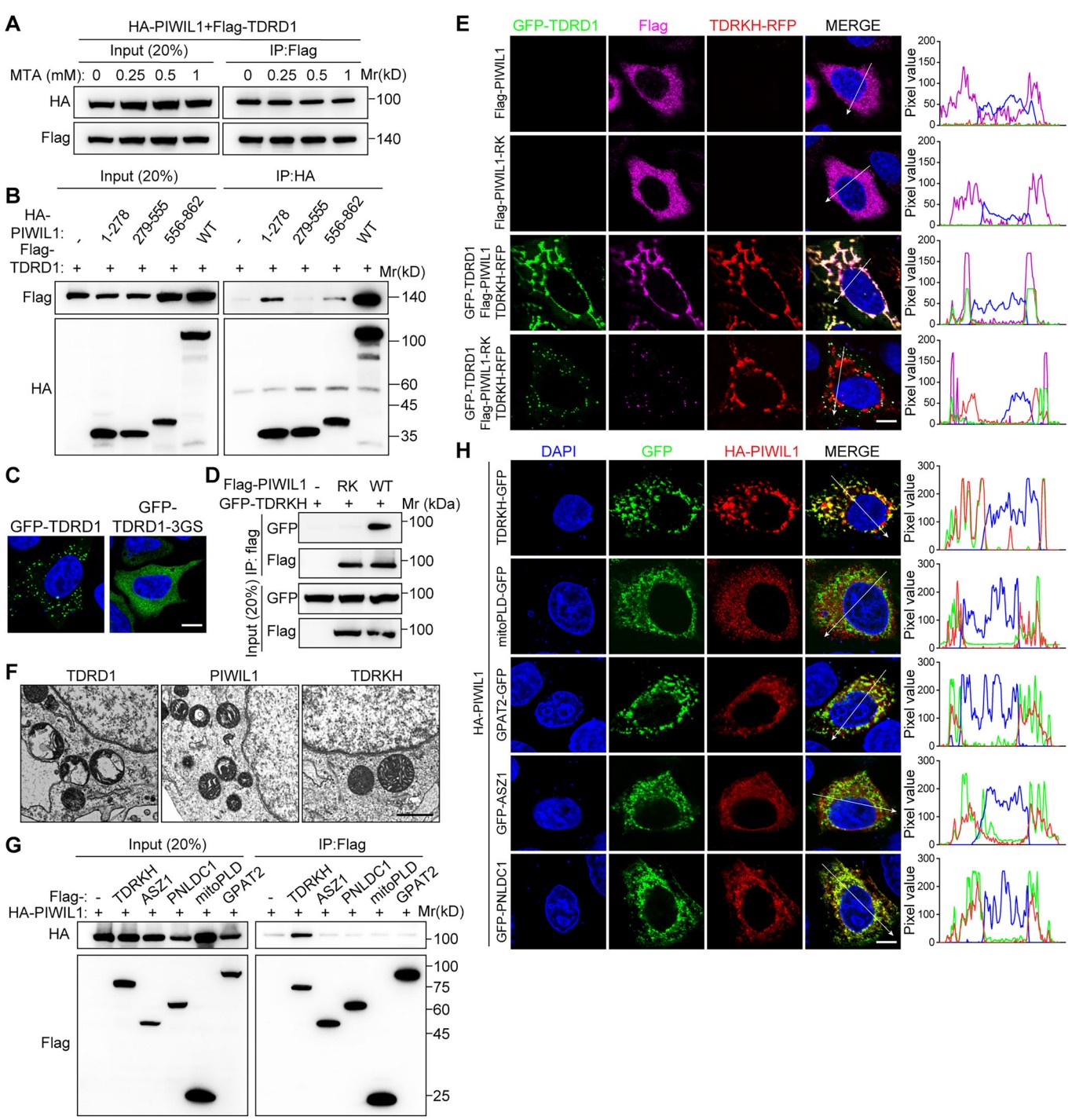

**Figure EV3. TDRKH-PIWIL1 complex cooperates with ASZ1-PIWIL2 complex to recruit TDRD1 to mitochondria.**

(A) Co-IP assay in transfected 293T cells with MTA treatment. Flag-tagged and HA-tagged proteins were detected by WB. (B) Co-IP assay in 293T cells transfected with indicated constructs. Flag-tagged and HA-tagged proteins were detected by WB. (C) Images of HeLa cells transfected with the indicated plasmids. Scale bars, 10 μm. (D) Co-IP assay in 293T cells transfected with indicated constructs. Flag-tagged and GFP-tagged proteins were detected by WB. (E) HeLa cells were transfected with indicated constructs. Immunostaining was performed using the Flag antibody. Scale bars, 10 μm. Fluorescence intensity through positions denoted by the white lines are shown in right. (F) Image of TEM on HeLa cells transfected with GFP-TDRD1, Flag-PIWIL1, or TDRKH-RFP, respectively. Scale bars, 1 μm. (G) Co-IP assay in 293T cells transfected with indicated constructs. Flag-tagged and HA-tagged proteins were detected by WB. (H) HeLa cells were transfected with indicated constructs. Immunostaining was performed using the HA antibody. Scale bars, 10 μm. Fluorescence intensity through positions denoted by the white lines are shown in right.

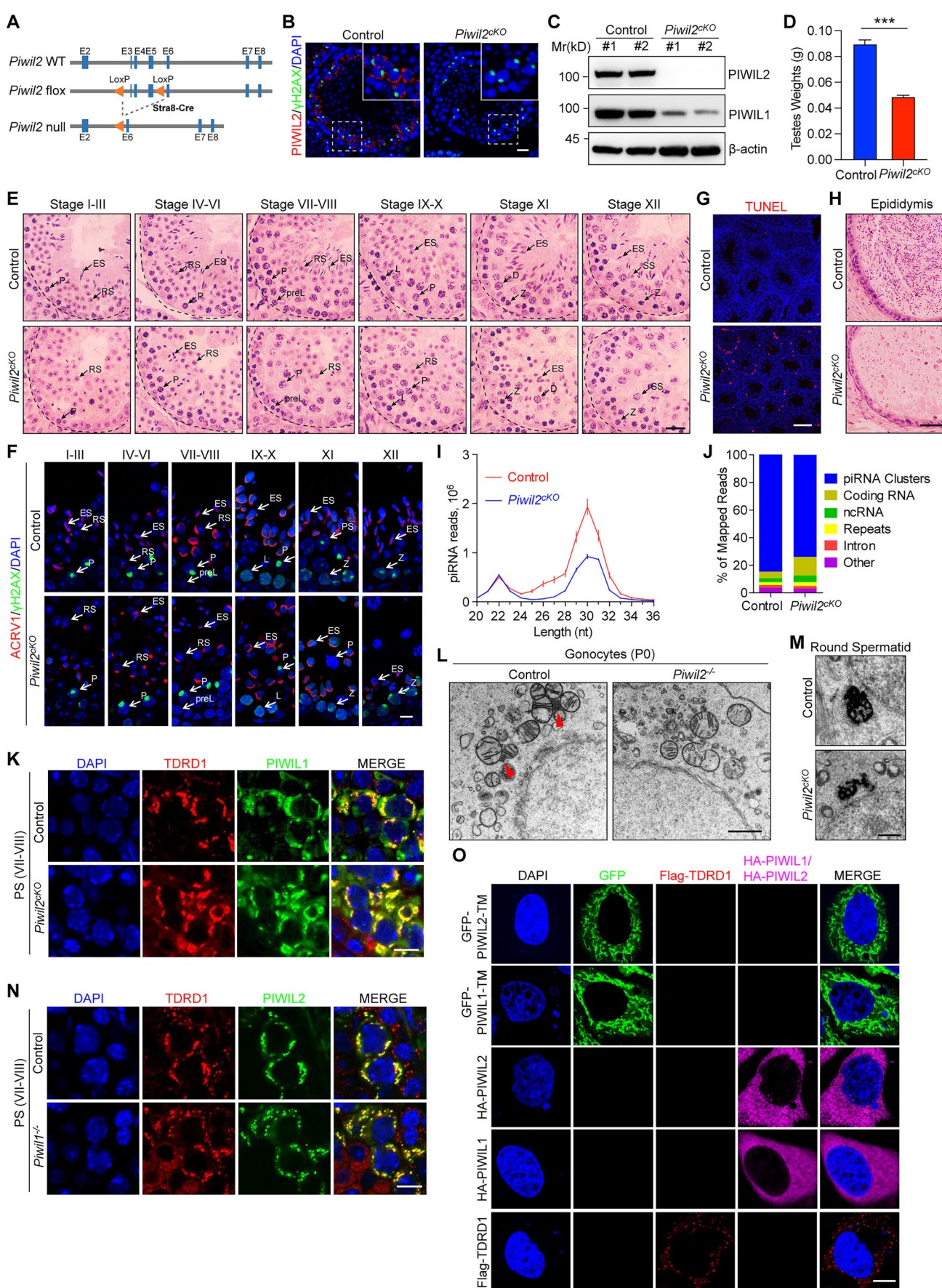

◀ **Figure EV4.  PIWIL2 is required for pachytene piRNA biogenesis.**

(A) A schematic diagram showing the gene targeting strategy for the generation of a *Piwil2* conditional allele. Cre-mediated deletion removed the exon 3–5 of *Piwil2* and generated a protein null allele. (B) Co-immunostaining of PIWIL2 and γH2AX on control and *Piwil2^cKO* adult testes. Scale bars, 20 μm. (C) WB of PIWIL2 and PIWIL1 expression in control and *Piwil2^cKO* adult testes. β-actin served as a control. (D) The average weight of adult testes ($n = 6$; ***$p < 0.001$). (E) H&E staining on adult testes. preL preleptotene, L leptotene, Z zygotene, P pachytene, D diplotene, RS round spermatid, ES elongated spermatid, SS secondary spermatocytes. Scale bars, 20 μm. (F) Co-immunostaining of ACRV1 and γH2AX on adult mouse testes. preL preleptotene, L leptotene, Z zygotene, P pachytene, D diplotene, M metaphase, RS round spermatid, ES elongated spermatid; Scale bars, 10 μm. (G) TUNEL assays on adult testes. Scale bars, 100 μm. (H) H&E staining of the mouse adult epididymis. Scale bars, 50 μm. (I) The length distribution of PIWIL1-piRNAs from adult testes. Data were normalized by total reads. $n = 2$. (J) Genomic annotation of PIWIL1-piRNAs from control and *Piwil2^cKO* adult testes. Data were representative of two biological replicates. (K) Co-immunostaining of TDRD1 and PIWIL1 on control and *Piwil2^cKO* adult testes. Stage VII–VIII seminiferous tubules were distinguished according to DAPI staining. PS pachytene spermatocytes. Scale bars, 10 μm. (L) Images of TEM on gonocytes from mouse P0 testes. IMCs were indicated by red arrowheads. Scale bars, 1 μm. (M) Images of TEM on round spermatids from adult testes showing CB structure. Scale bars, 1 μm. (N) Co-immunostaining of TDRD1 and PIWIL2 on control and *Piwil1^{-/-}* adult testes. Stage VII–VIII seminiferous tubules were distinguished according to DAPI staining. PS pachytene spermatocytes. Scale bars, 10 μm. (O) HeLa cells were transfected with indicated constructs. Immunostaining was performed using Flag, PIWIL1, and PIWIL2 antibodies. Scale bars, 10 μm. Data information: In (D, I), data were presented as mean ± s.e.m. In (D), $p$ values were calculated using Student's *t*-test (***$p < 0.001$). $p = 3.1469E\text{-}07$ (D).

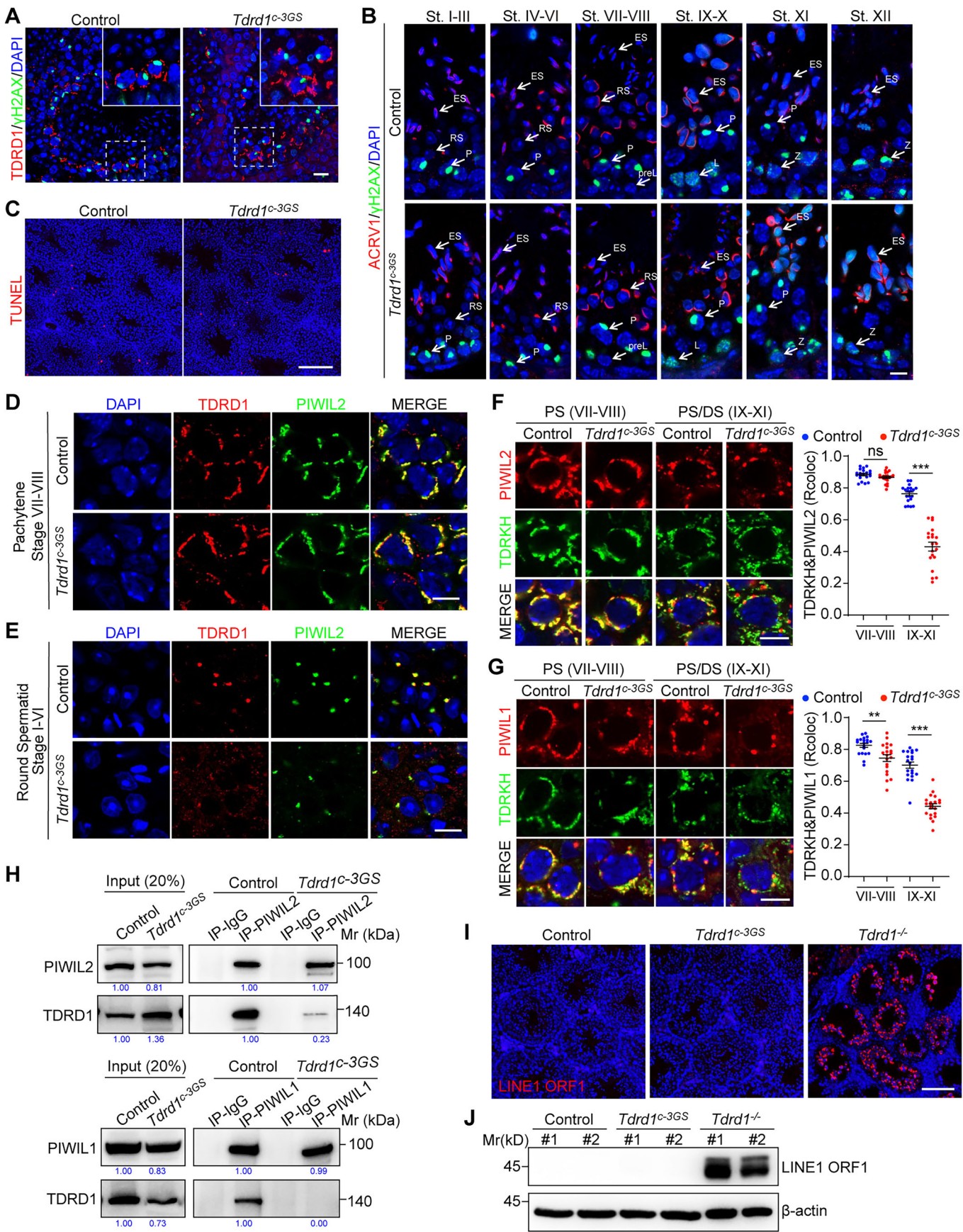

◀  **Figure EV5.  TDRD1 phase separation is required for IMC assembly, piRNA biogenesis, and spermiogenesis in adult testes.**

(A) Co-immunostaining of TDRD1 and γH2AX on control and *Tdrd1^{c-3GS}* adult testes. Scale bars, 20 μm. (B) Co-immunostaining of ACRV1 and γH2AX on adult mouse testes. preL preleptotene, L leptotene, Z zygotene, P pachytene, D diplotene, RS round spermatid, ES elongated spermatid. Scale bars, 10 μm. (C) TUNEL assays on adult testes. Scale bars, 200 μm. (D, E) Co-immunostaining of TDRD1 and PIWIL2 on pachytene spermatocytes (D) and round spermatids (E) from adult testes. Scale bars, 10 μm. (F, G) Co-immunostaining of PIWIL2-TDRKH (F) or PIWIL1-TDRKH (G) on adult testes. PS pachytene spermatocytes, DS diplotene spermatocytes. Scale bars, 10 μm. Quantification of the colocalization ratio is shown on the right. (*n* = 20; **p < 0.01; ***p < 0.001; ns not significant). The developmental stages of germ cells were distinguished according to DAPI staining. (H) Co-IP assay of PIWIL2 and PIWIL1 in control and *Tdrd1^{c-3GS}* adult testes. PIWIL2, PIWIL1, and TDRD1 protein levels were detected by WB. (I) Immunostaining of LINE1 ORF1 on adult testes. Scale bars, 50 μm. (J) WB of LINE1 ORF1 in adult testes. β-actin served as a control. Data information: In (F, G), data were presented as mean ± s.e.m and *p* values were calculated using Student's *t*-test (**p < 0.01; ***p < 0.001; ns not significant). *p* values from left to right (F): *p* = 0.1052, *p* = 2.1886E-13; *p* values from left to right (G): *p* = 0.0026, *p* = 2.6065E-12.

   