## [Peer Review File · The EMBO Journal]

PIWI proteins tether the piRNA biogenesis machinery to mitochondria during mammalian spermatogenesis

Jie Gao, Canmei Chen, Guanyi Shang, Wenyang Yu, Ting Zhao, Yunfang Zhang, Chen Chen, and Deqiang Ding

Corresponding author(s): Deqiang Ding (dingdeqiang@tongji.edu.cn)

Review Timeline:

Submission Date:	16th Apr 25
Editorial Decision:	28th May 25
Revision Received:	7th Jul 25
Editorial Decision:	14th Aug 25
Revision Received:	18th Aug 25
Accepted:	7th Sep 25

Editor: *Cornelius Schneider*

Transaction Report:

Dear Prof. Ding,

Thank you for submitting your manuscript for consideration by the EMBO Journal. It has now been seen by three referees whose comments are shown below.

Given the referees' positive recommendations, I would like to invite you to submit a revised version of the manuscript, addressing the comments of all three reviewers. I should add that it is EMBO Journal policy to allow only a single round of revision, and acceptance of your manuscript will therefore depend on the completeness of your responses in this revised version.

Thank you for the opportunity to consider your work for publication. I look forward to your revision.

Do not hesitate to contact me if there are any questions regarding the revision.

Yours sincerely,

Cornelius Schneider, PhD
Editor
The EMBO Journal
c.schneider@embojournal.org

Please remember: Digital image enhancement is acceptable practice, as long as it accurately represents the original data and

conforms to community standards. If a figure has been subjected to significant electronic manipulation, this must be noted in the figure legend or in the 'Materials and Methods' section. The editors reserve the right to request original versions of figures and the original images that were used to assemble the figure.

We realize that it is difficult to revise to a specific deadline. In the interest of protecting the conceptual advance provided by the work, we recommend a revision within 3 months (26th Aug 2025). Please discuss the revision progress ahead of this time with the editor if you require more time to complete the revisions. Use the link below to submit your revision:

Referee #1:

Review for: "PIWI proteins govern dynamics of piRNA biogenesis machinery in mice"

Summary:

This manuscript presents *in vitro* and *in vivo* data that suggest a hierarchy in the molecular assembly of some mammalian piRNA biogenesis factors on the surface of mitochondria. The authors provide valuable protein interaction screens for individual piRNA pathway proteins (ASZ1, MILI, TDRD1) (immunoprecipitation - mass spectrometry; IP-MS) from p20 mouse testes. Follow-up experiments test specific interactions between major components using microscopy, mouse genetics, and AlphaFold modeling of protein-protein interactions. This manuscript investigates an interesting topic, the assembly of piRNA processing complexes on the mitochondrial surface and in germ cell granules. Experiments are well designed and conducted and result in valuable results and data. However, the manuscript would benefit from some restructuring to improve flow, and the removal of less conclusive and only loosely related data to improve clarity and accessibility for a broader audience.

Major Points:

1. Reorganization of Results Section:

The manuscript would benefit from restructuring-separate *in vitro* and *in vivo* findings and follow the sequential assembly of ASZ1-MILI-TDRD1 and TDRKH-MIWI-TDRD1 complexes. This would clarify the narrative and better contextualize the final model.

2. piRNA Analyses - Figures 3C-G, 6K-N:

The interpretation of pachytene piRNA data is unclear and detracts from the main story. Claims about ping-pong amplification lack clarity and biological relevance, especially given the field's ongoing debate whether there is any ping-pong in pachytene piRNAs. The focus on L1-matching piRNAs overlooks the majority of pachytene piRNAs, which do not map to transposons. These analyses could be removed or has to be significantly revised.

3. LINE1 Expression - Figure 3H:

The conclusion that LINE1 activation contributes to the MILI^{CKO} phenotype is not supported by the data-only one tubule shows a signal. Alternative explanations (e.g., secondary upregulation due to germ cell damage) should be considered. Broader discussion on the elusive targets of pachytene piRNAs is warranted.

4. PPI Experimental Design:

Different plasmid/tag combinations across reciprocal IPs (e.g., 2C, EV2, EV4) result in inconsistent enrichment and background. Clarify experimental design and ensure consistent use of controls.

5. Localization and Mitochondrial Targeting:

It remains unclear whether large fluorescent tags impact localization. For example, mitoPLD appears cytoplasmic in some images. Co-staining with mitochondrial markers is recommended to validate subcellular localization claims. It would be also helpful to clarify whether fluorescence images reflect live cells, native signal, or antibody staining.

6. Immunoprecipitation Labeling:

Specify input percentages used for IPs to help interpret enrichment.

7. AlphaFold Modeling:

Please provide modeling parameters, including seeds and confidence metrics (iPTM/PTM scores), for reproducibility.

8. MIWI and MILI Functions in Chromatoid Body (CB):

The evidence supporting a critical role for MIWI or MILI in CB formation is relatively limited, with only two studies cited, published 15 years apart. The authors may consider presenting this as a hypothesis or suggestion rather than stating it as an established scientific fact.

9. TEM Controls - Figure 1H:

While transfection of all components induces formation of electron dense structures between mitochondria (inter-mitochondrial-cement, IMC), controls using single components are missing. Prior reports show ASZ1 alone induces IMC-like structures-this should be explored for individual components for completeness.

10. Controls for Figures 4I-J:

The conclusions regarding TDRD1 localization in Mili/MIWI KOs require additional controls, including MIWI-TM constructs and co-staining across genotypes. These experiments could substantiate key claims about tethering and localization dynamics.

11. Figure 8 - Model:

This important figure is not cited in the text and lacks a legend. It should be integrated into the results or discussion with appropriate commentary. The preferences of MILI versus MIWI seem to be based mostly on in vitro data. This limitation should be discussed.

Minor Points:

- Nomenclature: The manuscript refers to ASZ1 as GASZ, a term more commonly associated with the *Drosophila* ortholog. For clarity, consider using standard nomenclature throughout. Similarly, to maintain consistency, MIWI2, MILI, and MIWI should be also identified as PIWIL4, PIWIL2, and PIWIL1, respectively. Additionally, it would be helpful to clarify whether 'IMC' and 'NUAGE' are used indistinguishable, and to clarify differences and commonalities between 'IMC', 'NUAGE', and 'CB' in the introduction.
- Writing and grammar: The manuscript would benefit from further language editing for grammatical clarity. For example, the word "interacting" on line 55 is tense-inconsistent, and the sentence on line 56 could be streamlined as: "However, the mechanism of piRNA biogenesis machinery assembly remains largely unknown." Also, consider revising line 41 to avoid redundancy with the following sentence.
- Relevance to human biology: The effort in generating multiple mouse models could be strengthened by referencing human fertility studies implicating the piRNA pathway and comparing relevant mutant phenotypes.

Referee #2:

Using heterologous expression approaches and mouse genetics, the authors reveal specific mouse PIWI-associated complexes and their roles in the formation of intermitochondrial cement (IMC) and piRNA biogenesis. This manuscript is results of diligent work and technically quite good. Studies of this sort are also good references and resources for further comparisons. However, below we list several areas that in our opinion warrant improvement. We also suggest additional analyses that, if fruitful, may elevate the biological significance of the study.

1. The title is rather vague and looks like a title of a review article on PIWI-piRNA pathways.
2. Figure 2 F-J: The results shown in these figures do not show that amounts of piRNAs loaded onto MILI after RNase A treatment are reduced. In addition, it is known that lack of piRNA loading destabilizes PIWI proteins. The authors should show levels of MILI protein (with loading control) and its associated piRNAs after RNase A treatment. It is also well known that RNase A treatment of cell lysate often results in the formation of nonspecific protein aggregation. The authors should also consider quantifying the results (protein bands) of several experiments.
3. Figure 3 E-G and EV3G-K: The authors conclude that the secondary piRNA biogenesis is even more active in MilicKO adult testes (line 216). However, as shown in Figure 3B, levels of both MILI and MIWI are markedly reduced in MilicKO adult testes. This strongly suggests that levels of piRNAs are also markedly reduced. Thus, these analyses were very likely done with markedly reduced piRNAs. What do the results really tell us?
4. Figure 3 J-K: The authors conclude that MILI plays essential roles in pachytene piRNA biogenesis but is not required for TDRD1 recruitment and IMC formation in pachytene spermatocytes (line 240-242). The results seem at odds with those shown in Figure 1 G and H. The reader may get confused. The authors should provide with some explanation to avoid unnecessary confusion.
5. Figure 4I: With MILI-TM, the authors conclude that TDRD1 preferentially interacts with MILI, rather than MIWI. To corroborate this conclusion, the authors should consider generating MIWI-TM and performing experiments similar to those shown in Figure 4I. Is tethering to mitochondria a key (or sole) requirement for TDRD1-binding?
6. Figure 4L: The authors state that the protein levels of MILI and TDRD1 were up-regulated in *Miwi*^{-/-} adult testes, indicating the compensatory role of MILI in *Miwi*^{-/-} mice. What might be the compensatory role?
7. Figure 6 J-N: similar to the comment #3. As shown in Figure 6J, levels of piRNAs are markedly reduced in *Tdrd1* cKO. Thus, these analyses were very likely done with markedly reduced piRNAs. Are they meaningful? What do the results really tell us?

Referee #3:

piRNAs interact with PIWI proteins and play critical roles in germ cell development. The piRNA biogenesis occurs in intermitochondrial cement (IMC) in mammalian germ cells. The mechanisms how IMC components associate with mitochondria is still incompletely understood. In the manuscript by Gao et al, the authors revealed that PIWI proteins, including MIWI and MILI, plays pivotal roles in governing the spatiotemporal dynamics of piRNA biogenesis machinery in mice. Using a set of experiments including in cell and in vivo approaches, they demonstrated MILI were recruited into IMCs by directly interacting with GASZ. Sequentially, piRNAs competitively bind MILI, leading to MILI dissociation with GASZ. Through systematic analysis of the interactions among piRNA pathway proteins, the authors identified two seed complex, GASZ-MILI-TDRD1 and TDRKH-MIWI-TDRD1, which play central roles in the assembly of piRNA biogenesis machinery. Overall, this is an interesting story that brings new information about how IMC formation and association with mitochondria. While the majority of the manuscript is persuasive, I only have a few suggestions for further improvement.

1. The authors use HEK293T cells for all biochemical interaction assays, while fluorescent imaging experiments are performed exclusively in HeLa cells. Given the distinct origins and characteristics of these two cell lines, please clarify the rationale behind selecting these two cell types for complementary assays, and discuss whether the biochemical interactions observed in HEK293T cells are representative of the localization patterns seen in HeLa cells.
2. Fig 2E: It is unclear whether GASZ Δ 1-32 affects its mitochondrial localization. Please perform co-staining with a mitochondrial marker (e.g., TOM20, MitoTracker) to confirm whether GASZ Δ 1-32 still localizes to mitochondria.
3. Fig 4D: It shows that all MIWI truncations retain interaction with TDRD1 based on co-IP, but it remains unverified in other systems. Please perform co-transfection experiments in HeLa cells to confirm the interaction.
4. Fig 4E: The authors propose that the TDRD1-MIWI interaction is independent of Tudor domain-arginine methylation recognition but enhanced by TDRD1 phase separation. However, this conclusion lacks direct experimental support. While the contribution of the phase separation domain has been experimentally tested, the role of the Tudor domains remains unaddressed. Please provide additional evidence mapping the interaction domains between TDRD1 and MILI/MIWI. These experiments are essential to support the proposed mechanism and distinguish LLPS-driven interactions from canonical Tudor-methylarginine binding.
5. Fig 6D-E: Tdrd1cKO mice resulted in a reduction in piRNA abundance, which would cause MIWI and MILI to remain near the mitochondria due to failed piRNA loading. However, MIWI-TDRKH and MILI-TDRKH co-localization is reduced. Please clarify why their co-localization is decreased in this context.
6. Fig 7: In the Tdrd1c-3GS mutant mice, is the interaction between TDRD1 and MILI/MIWI reduced?
7. Do the Tdrd1c-3GS mutant mice exhibit similar phenotypes to those observed in the Tdrd1cKO mice, as shown in Fig 6D-E?
8. Lines 67-76: The current consensus in the field classifies piRNAs into three types: fetal piRNAs, pre-pachytene piRNAs, and pachytene piRNAs. However, the manuscript mentions only fetal and pachytene piRNAs. Please clarify and update the classification accordingly.

Below is the point-by-point response to comments from the referees.

Referee #1:

Summary: This manuscript presents *in vitro* and *in vivo* data that suggest a hierarchy in the molecular assembly of some mammalian piRNA biogenesis factors on the surface of mitochondria. The authors provide valuable protein interaction screens for individual piRNA pathway proteins (ASZ1, MILI, TDRD1) (immunoprecipitation-mass spectrometry; IP-MS) from p20 mouse testes. Follow-up experiments test specific interactions between major components using microscopy, mouse genetics, and AlphaFold modeling of protein-protein interactions. This manuscript investigates an interesting topic, the assembly of piRNA processing complexes on the mitochondrial surface and in germ cell granules. Experiments are well designed and conducted and result in valuable results and data. However, the manuscript would benefit from some restructuring to improve flow, and the removal of less conclusive and only loosely related data to improve clarity and accessibility for a broader audience.

We sincerely appreciate the Referee's positive feedback and valuable recognition of our work. We are also very grateful for the excellent and constructive suggestions provided. We have carefully considered all comments and revised the manuscript accordingly.

Major Points:

1. Reorganization of Results Section: The manuscript would benefit from restructuring-separate *in vitro* and *in vivo* findings and follow the sequential assembly of ASZ1-MILI-TDRD1 and TDRKH-MIWI-TDRD1 complexes. This would clarify the narrative and better contextualize the final model.

Response: We thank the Referee for this valuable suggestion. We have separated *in vitro* and *in vivo* data and followed the sequential assembly of ASZ1-MILI-TDRD1 and TDRKH-MIWI-TDRD1 complexes to make the manuscript smooth and easy to understand.

2. piRNA Analyses - Figures 3C-G, 6K-N: The interpretation of pachytene piRNA data is unclear and detracts from the main story. Claims about ping-pong amplification lack clarity and biological relevance, especially given the field's ongoing debate whether there is any ping-pong in pachytene piRNAs. The focus on L1-matching piRNAs overlooks the majority of pachytene piRNAs, which do not map to transposons. These analyses could be removed or has to be significantly revised.

Response: We thank the Referee for this insightful suggestion. We agreed that the majority of pachytene piRNAs are produced through phased primary piRNA biogenesis pathway and the ping-pong amplification, if any, is not prominent. The LI-derived piRNAs account for only a very small portion of the pachytene piRNA population (Ding *et al*, 2018; Fu & Wang, 2014; Gou *et al*, 2014; Li *et al*, 2013;

Vourekas *et al.*, 2012). According to the Referee's suggestion, we removed the piRNA data about ping-pong amplification and L1-derived piRNAs from Fig 3, Fig 6, and Fig 7.

3. LINE1 Expression - Figure 3H: The conclusion that LINE1 activation contributes to the MILI cKO phenotype is not supported by the data-only one tubule shows a signal. Alternative explanations (e.g., secondary upregulation due to germ cell damage) should be considered. Broader discussion on the elusive targets of pachytene piRNAs is warranted.

Response: We thank the Referee for this comment. We observed that only a few of seminiferous tubules showed LINE1 positive signal in *Mili*^{cKO} testes. The majority of *Mili*^{cKO} germ cells didn't present LINE1 signal in our experiments. We cannot rule out that the cell damage in *Mili*^{cKO} testes caused LINE1 upregulation in a few of germ cells. In addition, the LI-derived piRNAs account for only a very small portion of the pachytene piRNA population (Ding *et al.*, 2018; Fu & Wang, 2014; Gou *et al.*, 2014; Li *et al.*, 2013; Vourekas *et al.*, 2012). To avoid detracting from the major conclusions of this manuscript, we have removed this part from the revised manuscript.

4. PPI Experimental Design: Different plasmid/tag combinations across reciprocal IPs (e.g., 2C, EV2, EV4) result in inconsistent enrichment and background. Clarify experimental design and ensure consistent use of controls.

Response: We thank the Referee for this comment. In this work, we used three tags for co-IP assay: Flag, HA, and GFP, all of which are widely used for overexpression in cells. We observed that different tagged recombinant proteins showed very different expression efficiency in cells, especially for some truncated proteins. When we designed the experiments, we chose different tag combinations to make sure the distinct recombinant proteins showed comparable expression level in the same experiment. We agreed that different tag combinations in co-IP may result in different enrichment and background. We always used the corresponding controls for each co-IP assay to reduce the interference from different tags in co-IP assay.

5. Localization and Mitochondrial Targeting: It remains unclear whether large fluorescent tags impact localization. For example, mitoPLD appears cytoplasmic in some images. Co-staining with mitochondrial markers is recommended to validate subcellular localization claims. It would be also helpful to clarify whether fluorescence images reflect live cells, native signal, or antibody staining.

Response: We thank the Referee for this suggestion. We performed the co-staining of mitoTracker with the tagged mitochondrial proteins which were used in recruitment screening assay in Figure 5 and Appendix Fig.S1. We have presented the data in revised Appendix Fig.S1A. The results showed that all of the tagged mitochondrial proteins, including ASZ1, TDRKH, mitoPLD and GPAT2 showed strong mitochondrial localization. Interestingly, we indeed observed mild cytoplasmic localization in some cells which may be caused by protein overexpression, transmembrane domain activity, or protein tags. However, such slight cytoplasmic

distribution does not affect our conclusion in this manuscript.

Appendix Fig.S1

6. Immunoprecipitation Labeling: Specify input percentages used for IPs to help interpret enrichment.

Response: We thank the Referee for this suggestion. Accordingly, we have added the input percentages in revised Fig 1, Fig 2, Fig 3, Fig 4, Fig EV1, Fig EV2, Fig EV3, and Fig EV5.

7. AlphaFold Modeling: Please provide modeling parameters, including seeds and confidence metrics (iPTM/PTM scores), for reproducibility.

Response: Per Referee's suggestion, we have added modeling parameters including random seed, iPTM scores, and PTM scores for AlphaFold Modeling in Methods and Protocols.

8. MIWI and MILI Functions in Chromatoid Body (CB): The evidence supporting a critical role for MIWI or MILI in CB formation is relatively limited, with only two studies cited, published 15 years apart. The authors may consider presenting this as a hypothesis or suggestion rather than stating it as an established scientific fact.

Response: We thank the Referee for this comment. Based on the published papers from our lab and other groups, MIWI and MILI are enriched in CB in round spermatids. The piRNA producing deficiency results in the fragmentation of CB structure (Ding *et al*, 2019; Ding *et al*, 2017; Reuter *et al*, 2011; Wei *et al*, 2024; Yabuta *et al*, 2011). However, we agreed that the precise roles of MILI and MIWI, as well as piRNAs, in CB formation are largely unknown. We cited more reference about CB formation in the introduction and added the statement "The precise role of PIWI-piRNA complexes in CB formation require further investigation" in the discussion".

9. TEM Controls - Figure 1H: While transfection of all components induces formation of electron dense structures between mitochondria (inter-mitochondrial-cement, IMC), controls using single components are missing. Prior reports show ASZ1 alone induces IMC-like structures-this should be explored for individual components for completeness.

Response: We thank the Referee for this suggestion. We performed TEM on transfected HeLa cells with GASZ, MILI, and TDRD1 individually or together. The results showed that overexpression of single component failed to induce IMC-like structure in HeLa cells (revised Fig EV1G). ASZ1 is reported to promote mitochondrial clustering through self-interaction (Miao *et al*, 2024). We speculate that the mitochondrial clustering is required but not sufficient for IMC formation. Additionally, we also performed TEM on transfected HeLa cells with TDRKH, MIWI, and TDRD1 individually or together. The results showed that only co-expression of TDRKH, MIWI and TDRD1 was able to induce IMC-like structure formation in HeLa cells (revised Fig 3H and EV3F).

Figure 1H and Figure EV1G

Figure 3H and Figure EV3F

10. Controls for Figures 4I-J: The conclusions regarding TDRD1 localization in Mili/MIWI KOs require additional controls, including MIWI-TM constructs and co-staining across genotypes. These experiments could substantiate key claims about tethering and localization dynamics.

Response: We thank the Referee for this suggestion. Accordingly, we performed co-staining of MILI and TDRD1 in *Miwi*^{-/-} adult testes. The results showed that TDRD1 showed comparable colocalization with MILI in *Miwi*^{-/-} pachytene spermatocytes (revised Fig EV4N). Meanwhile, we performed co-staining of MIWI and TDRD1 in *Mili*^{CKO} adult testes. The results showed that TDRD1 showed comparable colocalization with MIWI in *Mili*^{CKO} pachytene spermatocytes (revised Fig EV4K). Together, these data further indicated that loss of MILI or MIWI does not significantly impair TDRD1 recruitment to mitochondria.

Figure EV4K and N

Per Referee's suggestion, we also generated GFP-MIWI-TM and performed the recruitment assay in HeLa cells using GFP-MIWI-TM and GFP-MILI-TM respectively. The results showed that Both GFP-MIWI-TM and GFP-MILI-TM successfully recruited TDRD1 to mitochondria. MIWI and MILI were not able to recruit each other to mitochondria. When we co-transfected cells with GFP-MIWI-TM, Flag-TDRD1, and HA-MILI, both TDRD1 and MILI were recruited to mitochondria, indicating that TDRD1-MILI interaction is strong enough for their mitochondrial localization. On the other hand, when we co-transfected cells with GFP-MILI-TM, Flag-TDRD1, and HA-MIWI, only TDRD1 were recruited to mitochondria. The majority of MIWI are distributed in the cytoplasm, indicating that TDRD1-MIWI interaction is not enough for MIWI mitochondrial localization. Together, these data suggests that MILI interacts with TDRD1 stronger than MIWI. The new data were added as revised Fig 4H and Fig EV4O. We also revised the manuscript for these results.

Figure 4H

11. Figure 8 - Model: This important figure is not cited in the text and lacks a legend. It should be integrated into the results or discussion with appropriate commentary. The preferences of MILI versus MIWI seem to be based mostly on in vitro data. This limitation should be discussed.

Response: We thank the Referee for this comment. We agree that evidence for the preferences of MILI versus MIWI is limited. Considering that TDRD1, MIWI and MILI are co-localized in both IMC (pachytene spermatocytes), CB precursor (late pachytene spermatocytes and diplotene spermatocytes) and CB (round spermatids), we revised the model to avoid highlighting the preferences of MILI versus MIWI for TDRD1. We cited the revised Fig 8 in the discussion and added the legend for Fig 8.

Minor Points:

Nomenclature: The manuscript refers to ASZ1 as GASZ, a term more commonly associated with the *Drosophila* ortholog. For clarity, consider using standard nomenclature throughout. Similarly, to maintain consistency, MIWI2, MILI, and MIWI should be also identified as PIWIL4, PIWIL2, and PIWIL1, respectively. Additionally, it would be helpful to clarify whether 'IMC' and 'NUAGE' are used indistinguishable, and to clarify differences and commonalities between 'IMC', 'NUAGE', and 'CB' in the introduction.

Response: We thank the Referee for this comment. Accordingly, we replaced GASZ, MIWI, MILI, and MIWI2 to ASZ1, PIWIL1, PIWIL2, and PIWIL4 respectively.

The term “nuage” most commonly refers to a type of germ granule found in nurse cells of *Drosophila* (fruit flies), located near the nucleus and involved in transposon silencing and piRNA biogenesis (Anand & Kai, 2012; Czech *et al*, 2018; Ge *et al*, 2019; Lim & Kai, 2007). Some published articles also refer to the IMC (intermitochondrial cement) and CB (chromatoid body) appearing during mammalian spermatogenesis as “nuage” (Chuma *et al*, 2006; Meikar *et al*, 2011; Pan *et al*, 2005; Watanabe *et al*, 2011; Yokota, 2012). We propose that the term “nuage” represents various types of germ granules, rather than denoting one specific granule type. To maintain consistency with the majority of published literature and to avoid conceptual confusion, we refer to the piRNA-associated germ granules assembled among mitochondria as the IMC (intermitochondrial cement), and we refer to the granule appearing near the nucleus in round spermatids as the CB (chromatoid body) in the manuscript.

Writing and grammar: The manuscript would benefit from further language editing for grammatical clarity. For example, the word "interacting" on line 55 is tense-inconsistent, and the sentence on line 56 could be streamlined as: "However, the mechanism of piRNA biogenesis machinery assembly remains largely unknown." Also, consider revising line 41 to avoid redundancy with the following sentence.

Response: We thank the Referee for this comment. We have changed “via directly interacting with TDRKH” to “via direct interaction with TDRKH”. We revised the sentences on line 56 and line 41 according to the Referee’s suggestion. In addition, we improved the language throughout the text. All changes were highlighted in the revised manuscript.

Relevance to human biology: The effort in generating multiple mouse models could be strengthened by referencing human fertility studies implicating the piRNA pathway and comparing relevant mutant phenotypes.

Response: We thank the Referee for this comment. Indeed, many human mutations on piRNA-related gene are reported to cause human male infertility. However, the pathological mechanism underlying these mutations are largely unknown. It would be very helpful for this field to identify more human piRNA-associated gene mutations and then generating the corresponding mutant mice to study the function and biogenesis of piRNAs. We will prioritize this suggestion and put effort to do this in our future works.

Referee #2:

Using heterologous expression approaches and mouse genetics, the authors reveal specific mouse PIWI-associated complexes and their roles in the formation of intermitochondrial cement (ICM) and piRNA biogenesis. This manuscript is results of diligent work and technically quite good. Studies of this sort are also good references

and resources for further comparisons. However, below we list several areas that in our opinion warrant improvement. We also suggest additional analyses that, if fruitful, may elevate the biological significance of the study.

We greatly appreciate the positive appraisal of the Referee on our work and the constructive suggestions. We have thus put major efforts in addressing the remaining concerns to make the story better, as detailed below.

1. The title is rather vague and looks like a title of a review article on PIWI-piRNA pathways.

Response: We thank the Referee for this comment. We revised the title to “PIWI proteins tether piRNA biogenesis machinery to mitochondria in mice”.

2. Figure 2 F-J: The results shown in these figures do not show that amounts of piRNAs loaded onto MILI after RNase A treatment are reduced. In addition, it is known that lack of piRNA loading destabilizes PIWI proteins. The authors should show levels of MILI protein (with loading control) and its associated piRNAs after RNase A treatment. It is also well known that RNase A treatment of cell lysate often results in the formation of nonspecific protein aggregation. The authors should also consider quantifying the results (protein bands) of several experiments.

Response: We thank the Referee for this excellent suggestion about controls. We have detected the amounts of piRNAs loaded onto MILI with or without RNase A treatment. The results revealed that MILI-piRNA population (around 26nt) was significantly reduced after RNase A treatment (Revised Fig EV2F). We also performed WB using β -actin antibody as a loading control for MILI protein level after RNase A treatment (Revised Fig 2G, Fig 2H, and Fig 2I). In addition, we have quantified the protein bands in Fig 2G, Fig 2H, and Fig 2I. The quantification results were shown in revised figures.

Figure EV2F

3. Figure 3 E-G and EV3G-K: The authors conclude that the secondary piRNA biogenesis is even more active in Mili cKO adult testes (line 216). However, as shown in Figure 3B, levels of both MILI and MIWI are markedly reduced in MilicKO adult testes. This strongly suggests that levels of piRNAs are also markedly reduced. Thus, these analyses were very likely done with markedly reduced piRNAs. What do the results really tell us?

Response: We thank the Referee for this comment. The majority of pachytene piRNAs are produced through phased primary piRNA biogenesis pathway and the secondary piRNA biogenesis is not as active as fetal piRNA stage. We examined secondary piRNA biogenesis/ping-pong amplification using L1-derived piRNAs in our work. However, we agreed that the L1-derived piRNAs account for only a very small portion of the pachytene piRNA population (Ding *et al.*, 2018; Fu & Wang, 2014; Gou *et al.*, 2014; Li *et al.*, 2013; Vourekas *et al.*, 2012). This may undermine the significance of the analysis of secondary piRNA biogenesis in adult mice. Considering that the Referee #1 raised similar concern, we removed the piRNA data about secondary piRNA biogenesis/ping-pong amplification and L1-derived piRNAs from Fig 3, Fig 6, and Fig 7.

4. Figure 3 J-K: The authors conclude that MILI plays essential roles in pachytene piRNA biogenesis but is not required for TDRD1 recruitment and IMC formation in pachytene spermatocytes (line 240-242). The results seem at odds with those shown in Figure 1 G and H. The reader may get confused. The authors should provide with some explanation to avoid unnecessary confusion.

Response: We thank the Referee for this valuable suggestion. TDRD1 was located at IMCs in *Mili*^{cKO} pachytene spermatocytes. We observed sufficient IMC formation in *Mili*^{cKO} pachytene spermatocytes. We proposed that TDRKH-MIWI-TDRD1 was able to trigger IMC formation and TDRD1 recruitment in *Mili*^{cKO} pachytene spermatocytes. To avoid misunderstanding, we revised the conclusion to “loss of PIWIL2 does not significantly disrupt TDRD1 recruitment and IMC formation” in the manuscript.

5. Figure 4I: With MILI-TM, the authors conclude that TDRD1 preferentially interacts with MILI, rather than MIWI. To corroborate this conclusion, the authors should consider generating MIWI-TM and performing experiments similar to those shown in Figure 4I. Is tethering to mitochondria a key (or sole) requirement for TDRD1-binding?

Response: Per Referee’s suggestion, we generated GFP-MIWI-TM and performed the recruitment assay using GFP-MIWI-TM and GFP-MILI-TM respectively. The results showed that Both GFP-MIWI-TM and GFP-MILI-TM successfully recruited TDRD1 to mitochondria. MIWI and MILI were not able to recruit each other to mitochondria. When we co-transfected cells with GFP-MIWI-TM, Flag-TDRD1, and HA-MILI, both TDRD1 and MILI were recruited to mitochondria, indicating that TDRD1-MILI interaction is strong enough for their mitochondrial localization. On the other hand, when we co-transfected cells with GFP-MILI-TM, Flag-TDRD1, and HA-MIWI, only TDRD1 were recruited to mitochondria. The majority of MIWI are distributed in the

cytoplasm, indicating that TDRD1-MIWI interaction is not enough for MIWI mitochondrial localization. Together, these data suggests that MILI interacts with TDRD1 stronger than MIWI. The new data were added as revised Fig 4H and Fig EV4O. We also revised the manuscript for these results.

Figure 4H

In addition, TDRD1 interacts with MILI or MIWI in transfected 293T cells, indicating that tethering to mitochondria is not a key requirement for TDRD1-binding with its partners.

6. Figure 4L: The authors state that the protein levels of MILI and TDRD1 were up-regulated in *Miwi*^{-/-} adult testes, indicating the compensatory role of MILI in *Miwi*^{-/-} mice. What might be the compensatory role?

Response: We thank the Referee for this comment. We indeed observed the up-regulation of TDRD1 and MILI protein level in *Miwi*^{-/-} adult testes. MIWI is one of the key components for pachytene piRNA biogenesis and function. Loss of MIWI cause dramatical reduce of pachytene piRNA biogenesis. We propose that the germ cells express more TDRD1 and MILI proteins to compensate the MIWI role in piRNA biogenesis and produce as much piRNA as possible to maintain germ cell

development in *Miwi*^{-/-} testes.

7. Figure 6 J-N: similar to the comment #3. As shown in Figure 6J, levels of piRNAs are markedly reduced in *Tdrd1cKO*. Thus, these analyses were very likely done with markedly reduced piRNAs. Are they meaningful? What do the results really tell us?

Response: We thank the Referee for this comment. As we mentioned in comment #3, we have removed the piRNA data about secondary piRNA biogenesis/ping-pong amplification and L1-derived piRNAs from Fig 3, Fig 6, and Fig 7.

Referee #3:

piRNAs interact with PIWI proteins and play critical roles in germ cell development. The piRNA biogenesis occurs in intermitochondrial cement (IMC) in mammalian germ cells. The mechanisms how IMC components associate with mitochondria is still incompletely understood. In the manuscript by Gao et al, the authors revealed that PIWI proteins, including MIWI and MILI, plays pivotal roles in governing the spatiotemporal dynamics of piRNA biogenesis machinery in mice. Using a set of experiments including in cell and in vivo approaches, they demonstrated MILI were recruited into IMCs by directly interacting with GASZ. Sequentially, piRNAs competitively bind MILI, leading to MILI dissociation with GASZ. Through systematic analysis of the interactions among piRNA pathway proteins, the authors identified two seed complex, GASZ-MILI-TDRD1 and TDRKH-MIWI-TDRD1, which play central roles in the assembly of piRNA biogenesis machinery. Overall, this is an interesting story that brings new information about how IMC formation and association with mitochondria. While the majority of the manuscript is persuasive, I only have a few suggestions for further improvement.

We thank the Referee for the positive comments and constructive suggestions. We have revised the paper based on all the valuable recommendations provided.

1. The authors use HEK293T cells for all biochemical interaction assays, while fluorescent imaging experiments are performed exclusively in HeLa cells. Given the distinct origins and characteristics of these two cell lines, please clarify the rationale behind selecting these two cell types for complementary assays, and discuss whether the biochemical interactions observed in HEK293T cells are representative of the localization patterns seen in HeLa cells.

Response: We thank the Referee for this comment. We performed co-IP assay in 293T cells based on the fact that 293T cells exhibit exceptional transfection efficiency, high protein expression levels, rapid growth and easy maintenance. In fact, the 293T cells are the most widely used cell line for protein-protein interaction assay. However, we avoid to use 293T cells to perform fluorescent imaging experiments due to its poor adherence and irregular morphology. Instead, we used HeLa cells to perform fluorescent imaging experiments because HeLa cells are well-adherent, show consistent epithelial-like morphology, and are widely used for imaging. We revealed that the results from protein-protein interaction in 293T cells and colocalization

imaging in HeLa cells are quite consistent in our manuscript.

2. Fig 2E: It is unclear whether GASZ Δ 1-32 affects its mitochondrial localization. Please perform co-staining with a mitochondrial marker (e.g., TOM20, MitoTracker) to confirm whether GASZ Δ 1-32 still localizes to mitochondria.

Response: Per Referee's suggestion, we performed co-staining with MitoTracker in transfected cells. The result confirmed that both GASZ and GASZ Δ 1-32 sufficiently localizes to mitochondria (Revised Fig EV2C).

Figure EV2C

3. Fig 4D: It shows that all MIWI truncations retain interaction with TDRD1 based on co-IP, but it remains unverified in other systems. Please perform co-transfection experiments in HeLa cells to confirm the interaction.

Response: We thank the Referee for this suggestion. Accordingly, we transfected GFP-TDRD1 and Flag tagged MIWI fragments to HeLa cells and performed immunostaining in transfected cells. TDRD1 recruits full length of MIWI to cytoplasmic condensates. However, TDRD1 was not able to efficiently recruits all of MIWI fragments including MIWI Δ 1-278, Δ 279-555, and Δ 556-862 to cytoplasmic condensates (appended Figure R1). This is consistent with the data that all MIWI fragments showed significant reduced interaction with TDRD1. These data further imply that MIWI interacts with TDRD1 through multiple interfaces.

Figure R1

4. Fig 4E: The authors propose that the TDRD1-MIWI interaction is independent of Tudor domain-arginine methylation recognition but enhanced by TDRD1 phase separation. However, this conclusion lacks direct experimental support. While the contribution of the phase separation domain has been experimentally tested, the role of the Tudor domains remains unaddressed. Please provide additional evidence mapping the interaction domains between TDRD1 and MILI/MIWI. These experiments are essential to support the proposed mechanism and distinguish LLPS-driven interactions from canonical Tudor-methylarginine binding.

Response: We thank the Referee for this suggestion. To address the detail of TDRD1-MIWI interaction, we performed a series of co-IP assay in 293T cells. The results showed that deletion of Tudor domains from TDRD1 disrupts TDRD1-MILI interaction but did not disrupt TDRD1-MIWI (appended Figure R2A and B). MTA treatment impaired TDRD1-TD1234 interaction with MILI (appended Figure R2C and D) but not with MIWI. These data indicate the different mechanism by which TDRD1 recognizes MIWI and MILI respectively.

Figure R2

5. Fig 6D-E: *Tdrd1* cKO mice resulted in a reduction in piRNA abundance, which would cause MIWI and MILI to remain near the mitochondria due to failed piRNA loading. However, MIWI-TDRKH and MILI-TDRKH co-localization is reduced. Please clarify why their co-localization is decreased in this context.

Response: We thank the Referee for this comment. The pachytene piRNA biogenesis was significantly impaired in *Tdrd1*^{cKO} adult testes. However, A considerable amount of piRNAs were still loaded onto MILI and MIWI proteins. Furthermore, loss of TDRD1 completely disrupts IMC formation in pachytene spermatocyte, leading to instability of piRNA biogenesis machinery. We speculate that MIWI and MILI were recruited onto outer mitochondrial membrane by TDRKH and ASZ1 in *Tdrd1*^{cKO} pachytene spermatocyte. However, the instability of piRNA biogenesis machinery causes the reduced co-localization of MIWI and MILI with mitochondria.

6. Fig 7: In the *Tdrd1c-3GS* mutant mice, is the interaction between TDRD1 and MILI/MIWI reduced?

Response: Per Referee's suggestion, we performed IP assay in WT and *Tdrd1*^{c-3GS} testes using MIWI and MILI antibody. The results showed that TDRD1-3GS mutant showed significant reduced interaction with PIWIL1 and PIWIL2 in testes (revised Fig EV5H). This result further highlights the critical role of TDRD1 phase separation in assembly of piRNA biogenesis machinery.

Figure EV5H

7. Do the *Tdrd1c-3GS* mutant mice exhibit similar phenotypes to those observed in the *Tdrd1cKO* mice, as shown in Fig 6D-E?

Response: Per Referee's suggestion, we performed co-staining of MIWI and TDRKH, MILI and TDRKH in *Tdrd1^{c-3GS}* mutant mice. The results showed similar phenotypes to those observed in the *Tdrd1^{cKO}* mice. We added the new data as revised Fig EV5F and Fig EV5G.

Figure EV5F and EV5G

8. Lines 67-76: The current consensus in the field classifies piRNAs into three types: fetal piRNAs, pre-pachytene piRNAs, and pachytene piRNAs. However, the manuscript mentions only fetal and pachytene piRNAs. Please clarify and update the classification accordingly.

Response: We thank the Referee for this suggestion. We have clarified fetal piRNAs, pre-pachytene piRNAs, and pachytene piRNAs accordingly in the introduction.

Reference:

Anand A, Kai T (2012) The tudor domain protein kumo is required to assemble the nuage and to generate germline piRNAs in Drosophila. *The EMBO journal* 31: 870-882

Chuma S, Hosokawa M, Kitamura K, Kasai S, Fujioka M, Hiyoshi M, Takamune K, Noce T, Nakatsuji N (2006) Tdrd1/Mtr-1, a tudor-related gene, is essential for male germ-cell differentiation and nuage/germinal granule formation in mice. *Proceedings of the National Academy of Sciences of the United States of America* 103: 15894-15899

Czech B, Munafo M, Ciabrelli F, Eastwood EL, Fabry MH, Kneuss E, Hannon GJ (2018) piRNA-Guided Genome Defense: From Biogenesis to Silencing. *Annu Rev Genet* 52: 131-157

Ding D, Liu J, Dong K, Melnick AF, Latham KE, Chen C (2019) Mitochondrial membrane-based initial separation of MIWI and MILI functions during pachytene piRNA biogenesis. *Nucleic acids research* 47: 2594-2608

Ding D, Liu J, Dong K, Midic U, Hess RA, Xie H, Demireva EY, Chen C (2017) PNLDC1 is essential for piRNA 3' end trimming and transposon silencing during spermatogenesis in mice. *Nat Commun* 8: 819

Ding D, Liu J, Midic U, Wu Y, Dong K, Melnick A, Latham KE, Chen C (2018) TDRD5 binds piRNA precursors and selectively enhances pachytene piRNA processing in mice. *Nature communications* 9: 1-13

Fu Q, Wang PJ (2014) Mammalian piRNAs: Biogenesis, function, and mysteries.

Spermatogenesis 4: e27889

Ge DT, Wang W, Tipping C, Gainetdinov I, Weng Z, Zamore PD (2019) The RNA-Binding

ATPase, Armitage, Couples piRNA Amplification in Nuage to Phased piRNA Production on

Mitochondria. *Mol Cell* 74: 982-995 e986

Gou L-T, Dai P, Yang J-H, Xue Y, Hu Y-P, Zhou Y, Kang J-Y, Wang X, Li H, Hua M-M (2014)

Pachytene piRNAs instruct massive mRNA elimination during late spermiogenesis. *Cell*

research 24: 680-700

Li XZG, Roy CK, Dong XJ, Bolcun-Filas E, Wang J, Han BW, Xu J, Moore MJ, Schimenti JC,

Weng ZP *et al* (2013) An Ancient Transcription Factor Initiates the Burst of piRNA Production

during Early Meiosis in Mouse Testes. *Molecular Cell* 50: 67-81

Lim AK, Kai T (2007) Unique germ-line organelle, nuage, functions to repress selfish genetic

elements in *Drosophila melanogaster*. *Proceedings of the National Academy of Sciences of*

the United States of America 104: 6714-6719

Meikar O, Da Ros M, Korhonen H, Kotaja N (2011) Chromatoid body and small RNAs in male

germ cells. *Reproduction* 142: 195-209

Miao J, Wang C, Chen W, Wang Y, Kakasani S, Wang Y (2024) GASZ self-interaction clusters

mitochondria into the intermitochondrial cement for proper germ cell development. *PNAS*

Nexus 3: pgad480

Pan J, Goodheart M, Chuma S, Nakatsuji N, Page DC, Wang PJ (2005) RNF17, a component

of the mammalian germ cell nuage, is essential for spermiogenesis. *Development* 132:

4029-4039

Reuter M, Berninger P, Chuma S, Shah H, Hosokawa M, Funaya C, Antony C, Sachidanandam R, Pillai RS (2011) Miwi catalysis is required for piRNA amplification-independent LINE1 transposon silencing. *Nature* 480: 264-267

Vourekas A, Zheng Q, Alexiou P, Maragkakis M, Kirino Y, Gregory BD, Mourelatos Z (2012) Mili and Miwi target RNA repertoire reveals piRNA biogenesis and function of Miwi in spermiogenesis. *Nature structural & molecular biology* 19: 773-781

Watanabe T, Chuma S, Yamamoto Y, Kuramochi-Miyagawa S, Totoki Y, Toyoda A, Hoki Y, Fujiyama A, Shibata T, Sado T *et al* (2011) MITOPLD is a mitochondrial protein essential for nuage formation and piRNA biogenesis in the mouse germline. *Dev Cell* 20: 364-375

Wei H, Gao J, Lin DH, Geng R, Liao J, Huang TY, Shang G, Jing J, Fan ZW, Pan D *et al* (2024) piRNA loading triggers MIWI translocation from the intermitochondrial cement to chromatoid body during mouse spermatogenesis. *Nat Commun* 15: 2343

Yabuta Y, Ohta H, Abe T, Kurimoto K, Chuma S, Saitou M (2011) TDRD5 is required for retrotransposon silencing, chromatoid body assembly, and spermiogenesis in mice. *The Journal of cell biology* 192: 781-795

Yokota S (2012) Nuage proteins: their localization in subcellular structures of spermatogenic cells as revealed by immunoelectron microscopy. *Histochem Cell Biol* 138: 1-11

Dear Prof. Ding,

Thank you for submitting a revised version of your manuscript. Your study has now been seen by all original referees, who find that their previous concerns have been addressed and now recommend publication of the manuscript. There remain only a few mainly editorial points that have to be addressed before I can extend formal acceptance of the manuscript:

- Please reduce the number of keywords on the abstract page to five (ideally choosing broad general terms).
- As we are switching from a free-text author contribution statement towards a more formal statement based on Contributor Role Taxonomy (CRediT) terms, please remove the present Author Contribution section and instead specify each author's contribution(s) directly in the Author Information page of our submission system during upload of the final manuscript. See <https://casrai.org/credit/> for more information.
- Please add page numbers throughout the file and in the table of contents on the title page of the APPENDIX FILE WITH ToC
- Please provide suggestions for a short 'blurb' text prefacing and summing up the conceptual aspect of the study in two sentences (max. 250 characters), followed by 3-5 one-sentence 'bullet points' with brief factual statements of key results of the paper; they will form the basis of an editor-written 'Synopsis' accompanying the online version of the article. Please also provide an altered synopsis image, making sure that the aspect ratio conforms to our website's format - it should be exactly 550 pixels wide and between 300-600 pixels high.
- Please remove the R&T table from the manuscript file and upload separately
- Please rename the "Materials and Methods" section to "Methods"
- Please provide the specific URL for PRJNA1219779 dataset in the data availability statement.
- Please note that the exact p values are not provided in the legends of figures 3B, 6A, C, D, E, L; 7F, H; EV4 D, EV5 F, G; S2 F.
- Please indicate the statistical test used for data analysis in the legends of figures 1C, D; 3A, B; EV5 F, G
- Please note that the error bars are not defined in the legends of figures 3B, EV5 F, G.

With best regards,
Cornelius Schneider

Cornelius Schneider, PhD
Editor | The EMBO Journal
c.schneider@embojournal.org

Please refer to our figure preparation guideline in order to ensure proper formatting and readability in print as well as on screen:

See also figure legend guidelines:

<https://www.embopress.org/page/journal/14602075/authorguide#figureformat>

Use the link below to submit your revision:

Referee #1:

The authors have done an excellent job in improving this manuscript and in answering all the reviewers' questions.

Referee #2:

On the whole it has been improved and the authors have addressed most of the reviewers concerns, mostly by actually experimentally addressing the reviewers concerns.

When the authors cite relevant papers in the text, refs are located after period/full stop [e.g., ----. (Gao et al., 2024)]. We normally put the period after citation [e.g., ---- (Gao et al., 2024)].

Referee #3:

The authors have addressed all my comments and suggestions and I support its publication without further revision.

Below is the point-by-point response to comments from the referees.

Referee #1:

The authors have done an excellent job in improving this manuscript and in answering all the reviewers' questions.

We are delighted that the Referee is satisfied with our revision, and greatly appreciate his/her constructive suggestions for improving our manuscript.

Referee #2:

On the whole it has been improved and the authors have addressed most of the reviewers' concerns, mostly by actually experimentally addressing the reviewers' concerns.

We greatly appreciate the Referee for the constructive suggestions.

When the authors cite relevant papers in the text, refs are located after period/full stop [e.g., ----. (Gao et al., 2024)]. We normally put the period after citation [e.g., ---- (Gao et al., 2024).].

Response: Per Referee's comment, we have put the punctuations after citations in manuscript.

Referee #3:

The authors have addressed all my comments and suggestions and I support its publication without further revision.

We are delighted that the Referee is satisfied with our revision, and sincerely thank his/her careful inspection of our manuscript.

Dear Prof. Ding,

I am pleased to inform you that your manuscript has been accepted for publication in the EMBO Journal.

Yours sincerely,

Cornelius Schneider, PhD
Editor
The EMBO Journal
c.schneider@embojournal.org
